# EgoNight: Towards Egocentric Vision Understanding at Night with a Challenging Benchmark

**Deheng Zhang**[1*], **Yuqian Fu**[1*†], **Runyi Yang**[1], **Yang Miao**[1], **Tianwen Qian**[2], **Xu Zheng**[1,3], **Guolei Sun**[4], **Ajad Chhatkuli**[1], **Xuanjing Huang**[5], **Yu-Gang Jiang**[5], **Luc Van Gool**[1], **Danda Pani Paudel**[1]

[1]INSAIT, Sofia University "St. Kliment Ohridski"    [2]East China Normal University
[3]HKUST(GZ)    [4]Nankai University    [5]Fudan University

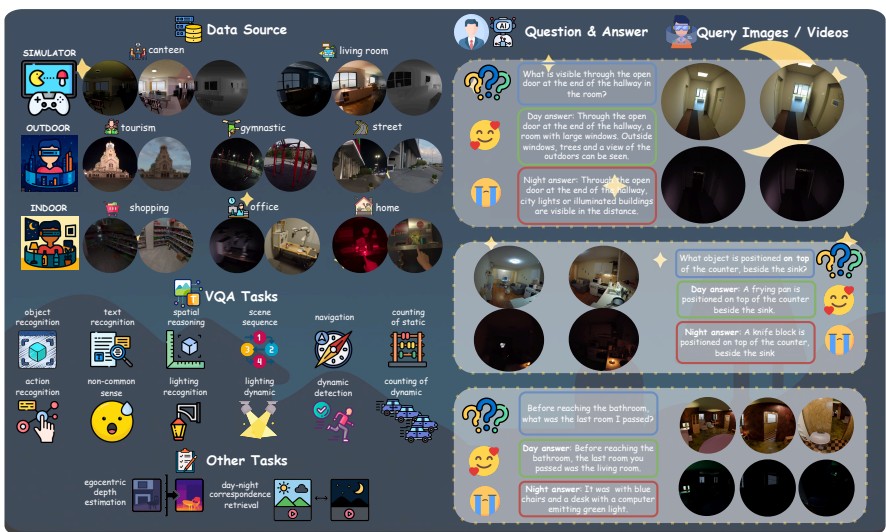

Figure 1: **Overview of the EgoNight.** EgoNight integrates diverse video sources spanning synthetic environments, real-world indoor and outdoor scenes, recorded under both daytime and nighttime conditions, with spatial and temporal alignment. It consists of three benchmarks: (i) *egocentric VQA* as the primary focus, (ii) *day–night correspondence retrieval*, and (iii) *egocentric depth estimation*, all targeting the challenges of low-light egocentric vision. The day–night alignment (illustrated on the right with VQA examples) enables rigorous analysis of illumination gaps in MLLMs.

## ABSTRACT

Most egocentric vision benchmarks focus on daytime scenarios, overlooking the low-light conditions common in real-world applications. We introduce *EgoNight*, the first comprehensive benchmark for nighttime egocentric vision, centered on visual question answering (VQA). EgoNight features day–night aligned videos, including both Blender-rendered synthetic data and real-world recordings, enabling higher-quality night annotations and direct comparison across illumination conditions. Using these paired videos, we construct *EgoNight-VQA* with a day-augmented night auto-labeling engine, extensive human verification, and double-checked QA pairs. The dataset contains 3658 QA pairs across 90 videos, covering 12 QA types and requiring over 300 hours of annotation. Evaluations of state-of-the-art multimodal large language models reveal substantial performance drops from day to night, highlighting the difficulty of reasoning under low-light conditions. EgoNight also includes two auxiliary tasks: day–night correspondence retrieval and nighttime egocentric depth estimation. We hope EgoNight provides a strong foundation for illumination-robust egocentric vision research. Code and data are available at https://dehezhang2.github.io/EgoNight/.

---

* means equal contribution; † denotes corresponding author.

# 1 INTRODUCTION

With the rapid development of wearable devices, egocentric vision understanding has become increasingly important. Unlike third-person vision, egocentric perception naturally aligns with the way humans perceive, understand, and interact with the world. A robust egocentric vision system can not only serve as an intelligent assistant in daily activities Yang et al. (2025) but also play a crucial role in embodied AI and robotic learning Li et al. (2025a); Kareer et al. (2025). Beyond these general applications, egocentric vision holds unique potential for assisting specific user groups such as people who are blind or visually impaired Xiao et al. (2025), or physically disabled Zhang et al. (2023a), enabling technologies that enhance navigation, accessibility, and real-time scene understanding. Significant efforts have been made to advance egocentric vision understanding, including the construction of large-scale ego-centric datasets such as EPIC-KITCHENS Damen et al. (2020), Ego4D Grauman et al. (2022), and Ego-Exo4D Grauman et al. (2024); the design of diverse and challenging benchmarks such as EgoTaskQA Jia et al. (2022), EgoSchema Mangalam et al. (2023), and EgoTempo Plizzari et al. (2025); and the development of egocentric multimodal large language models (MLLMs) such as EgoVLPv2 Pramanick et al. (2023), EgoGPT Yang et al. (2025), and Exo2Ego Zhang et al. (2025a). Despite these advances, almost all prior works focus on daytime scenarios with favorable lighting. In contrast, real-world egocentric systems, for example, intelligent personal assistants for navigation, must operate at night, under low light, uneven illumination, and severely limited visibility. This motivates us to investigate egocentric vision at night, focusing on complex scene understanding and reasoning tasks.

A key challenge in building such a benchmark is obtaining nighttime videos that reflect real-world conditions while ensuring high-quality annotations. To this end, we emphasize *day–night aligned videos*, which allow daytime data to support nighttime annotation and enable direct performance comparisons across lighting conditions. Since perfectly aligned real-world day–night pairs are difficult to collect, we first use Blender Iraci (2013) to precisely control scene layouts, camera trajectories, and lighting. This yields **EgoNight-Synthetic**, containing 50 ideally aligned egocentric day–night pairs across diverse indoor scenes and illumination levels. To complement synthetic data, we design a *video-guided recording protocol* to build **EgoNight-Sofia**, comprising 20 spatially and temporally aligned real-world day–night pairs from indoor and outdoor scenarios with varied light sources, including streetlights, flashlights, and candles. Finally, we include 20 nighttime videos from the Oxford Day-and-Night dataset Wang et al. (2025b), termed **EgoNight-Oxford**, as an additional unaligned testbed. Together, these three sources form the **EgoNight** dataset, summarized in Fig. 1.

The videos in EgoNight enable the construction of challenging benchmarks for evaluating existing models. We focus primarily on egocentric video question answering (VQA), a flagship task for assessing high-level egocentric understanding. To comprehensively evaluate model capabilities, we define a diverse set of QA types covering both well-studied tasks and underexplored dimensions, further grouped into paired and unpaired categories depending on whether day–night counterparts share the same questions and answers. To build the benchmark at scale, we develop a novel three-stage *day-augmented auto-labeling pipeline* that leverages daytime videos to generate QA pairs for nighttime clips, followed by extensive human verification. Constructing EgoNight and annotating VQA required **over 300 hours of human effort**, with each QA pair checked by at least one expert annotator. This results in **EgoNight-VQA**, a high-quality dataset containing 3,658 QA pairs. Beyond VQA, we introduce two auxiliary testbeds: day–night correspondence retrieval, which evaluates cross-illumination matching, and nighttime egocentric depth estimation, which is essential for navigation and embodied interaction. Extensive experiments across three video sources, three tasks, and 10 state-of-the-art MLLMs show that nearly all models, including closed-source models such as GPT and Gemini, struggle on EgoNight, with a clear and consistent gap between daytime and nighttime performance. Moreover, our newly introduced QA types, including lighting recognition/dynamics, scene sequence reasoning, navigation, and non-common-sense reasoning, are substantially more challenging than standard categories. We further find that synthetic data correlates strongly with real data and improves real-world performance through fine-tuning. Pilot studies suggest that such fine-tuning helps adapt the vision encoder to low-light domains and align the language model with uncertain nighttime visual features.

Our main contributions are threefold: i) **EgoNight Dataset**: We present the first egocentric dataset that systematically addresses nighttime conditions, featuring day–night aligned videos from synthetic (EgoNight-Synthetic), real-world (EgoNight-Sofia), and existing (EgoNight-Oxford) sources.

ii) **Benchmark Suite**: We build a comprehensive benchmark centered on egocentric VQA with diverse QA types and 3658 fully human-verified QA pairs, complemented by egocentric depth estimation at night and day–night correspondence retrieval tasks. iii) **Empirical Insights**: Extensive evaluations reveal clear day–night performance gaps, underscoring illumination robustness as a key challenge; our newly proposed QA types are also validated to pose practical difficulties for current MLLMs.

## 2 RELATED WORKS

### 2.1 EGOCENTRIC DATASETS AND VQA BENCHMARKS

A series of large-scale egocentric datasets, such as EPIC-KITCHENS Damen et al. (2020), Ego4D Grauman et al. (2022), Ego-Exo4D Grauman et al. (2024), and EgoExoLearn Huang et al. (2024), have laid the foundation for a wide range of tasks, including action recognition Sudhakaran et al. (2019), object detection Ren & Gu (2010), pose estimation Luo et al. (2021), video generation Liu et al. (2021), Ego-Exo correspondence Fu et al. (2025); Pan et al. (2025) and Ego-Exo translation Liu et al. (2024); Mahdi et al. (2025). Among these, we are particularly interested in egocentric visual question answering (VQA) Fan (2019), which provides a natural and human-like framework for comprehensively evaluating model performance through question–answer interactions. In recent years, several egocentric VQA benchmarks have been proposed, including EgoVQA Fan (2019), EgoTaskQA Jia et al. (2022), EgoSchema Mangalam et al. (2023), EgoThink Cheng et al. (2024), EgoTempo Plizzari et al. (2025), EgoCross Li et al. (2025b), EgoBlind Xiao et al. (2025), EgoMemoria Ye et al. (2024), HourVideo Chandrasegaran et al. (2024), EgoLifeQA Yang et al. (2025) with different focuses. However, nearly all of them are confined to daytime or well-lit scenarios, leaving model performance in low-light or nighttime conditions largely unexplored. The Oxford Day-and-Night dataset Wang et al. (2025b) is a partial exception but was not designed for VQA and lacks day–night alignment. This makes EgoNight and EgoNight-VQA fundamentally distinct from prior benchmarks.

### 2.2 MLLMS FOR VIDEO UNDERSTANDING

The rapid development of multimodal large language models (MLLMs) has substantially advanced the frontier of video understanding. Prominent open-source models include Qwen-VL Bai et al. (2023), InternVL Chen et al. (2024b), Video-LLaMA Zhang et al. (2023b), LLaVA-NeXT-Video Li et al. (2024), and GLM-V Hong et al. (2025), while closed-source commercial systems such as GPT-4V Achiam et al. (2023) and Gemini Comanici et al. (2025) demonstrate even stronger capabilities in video captioning, summarization, and open-ended visual question answering. Building on these advances, egocentric MLLMs have emerged to adapt foundation models from exocentric to first-person perspectives. Representative examples include EgoVLPv2 Pramanick et al. (2023) for improved video–language cross-modal fusion, EgoGPT Yang et al. (2025) fine-tuned with egocentric captioning and QA, MM-Ego Ye et al. (2024) with a memory mechanism for long videos, and Exo2Ego Zhang et al. (2025a) leveraging exocentric data for egocentric generalization. These works highlight the potential of MLLMs as egocentric assistants. However, nearly all of them are developed and tested under well-lit daytime conditions, leaving their robustness in low-light or nighttime scenarios unexplored. Nevertheless, almost all existing MLLMs are developed and evaluated under well-lit daytime conditions, with little consideration of low-light or nighttime videos, leaving their robustness in low-light or nighttime scenarios unexplored.

### 2.3 CROSS-DOMAIN GENERALIZATION

Domain generalization Zhou et al. (2022) is a long-standing challenge in computer vision, where models trained on one distribution must adapt to another. Shifts can arise from semantic drift, style changes, or variations in weather and lighting. Many algorithms have been validated across tasks such as image classification Li et al. (2017); Zhou et al. (2021); Huang et al. (2023); Peng et al. (2024; 2026), object detection Fu et al. (2024); Li et al. (2025c), action recognition Pan et al. (2020); Fu et al. (2020); Bian et al. (2011); Zou et al. (2021; 2020), few-shot learning Guo et al. (2020); Fu et al. (2021; 2023); Zhang et al. (2025c), and autonomous driving Li et al. (2022; 2023a). In contrast, cross-domain transfer for MLLMs, especially in video understanding, remains underexplored,

with only a few recent attempts (e.g., CL-CrossVQA Zhang et al. (2025b), VQA-GEN Unni et al. (2023), Super-CLEVR Li et al. (2023c)). However, none of them are targeted for egocentric video, which is naturally different from exocentric videos in terms of recorded images, camera motion, and contained information. The most relevant effort to us is EgoCross Li et al. (2025b), an egocentric VQA benchmark that moves beyond daily activities to evaluate model generalization across distinct long-tail, specialized domains such as surgery and industrial settings. In this paper, however, we investigate MLLMs from a different perspective, robustness under nighttime conditions, a common and ubiquitous scenario in daily life, yet previously overlooked dimension of domain generalization in egocentric video understanding.

## 3 EGONIGHT DATASET & BENCHMARKS

### 3.1 VIDEO SOURCE COLLECTION

**Overview & Design Principles.** EgoNight is built to systematically evaluate MLLMs under challenging nighttime conditions, which are critical for developing robust intelligent assistants. The collection of video sources follows four principles: ① *Reflect real-world challenges*, such as walking on dimly lit streets or navigating indoors during power outages; ② Involve *natural camera movements* and preferably capture *actions and interactions* with the environment; ③ Ensure *diversity of scenarios, illumination, and task difficulties*. ④ Enable rigorous analysis through *day–night paired videos*, where scenes, trajectories, and actions remain consistent across conditions so that differences can be attributed solely to illumination. To meet these requirements, EgoNight integrates three complementary video sources, as illustrated in the upper part of Fig. 2 and detailed below.

**EgoNight-Synthetic.** To obtain perfectly aligned day–night video pairs, we used a simulation environment where every element can be precisely controlled, including the scene layout, camera path, and lighting. This ensures that the day and night videos match exactly at the pixel and frame level, with lighting being the only difference. We first employ Infinigen Raistrick et al. (2023) to generate diverse indoor 3D scenes. Human annotators cleaned and refined these scenes, then simulated walking through the space at a normal speed (1.2 m/s), recording the camera trajectory. We replayed the same trajectory under different lighting conditions. Daytime videos were rendered using Blender Iraci (2013), and we adjusted the lighting to create the corresponding nighttime versions. In total, EgoNight-Synthetic contains 50 pairs of egocentric videos, covering more than 100 environment assets. These include indoor scenes such as kitchens, bathrooms, and living rooms, populated with over 50 diverse object categories (e.g., windows, tables, beds, chairs, lamps, bookshelves, plates). We design multiple illumination setups, ranging from uniformly lit rooms to sparsely localized lighting, and incorporate three difficulty levels with a different range of motion blur, sensor noise, and illumination level. Besides RGB frames, Blender also allows us to generate ground-truth depth and normals (see Appendix Sec. A.1), making EgoNight-Synthetic richer and more versatile.

**EgoNight-Sofia.** To include realistic human–environment interactions missing from synthetic videos, we also recorded our own day–night paired videos. Capturing perfectly aligned real-world pairs is challenging, so we designed a practical video-guided recording strategy with post-trimming for better alignment. We first record a daytime video with an ego-wearer exploring an environment while viewing the live camera feed on a phone screen. For the nighttime version, the same person, device, and viewpoint are kept unchanged. Instead of using live preview, the recorded daytime video is played back on the phone, serving as visual guidance to help the ego-wearer match walking speed, viewpoints, and actions. After brief practice, this approach proved more stable and reliable than methods like using landmarks or memorized trajectories. Post-trimming is applied to further refine spatial and temporal consistency. Our real-world dataset, EgoNight-Sofia, contains 20 day–night paired videos recorded in Sofia, Bulgaria. Despite its modest size, it is a rare resource capturing diverse real-world everyday scenarios, including apartments, offices, grocery stores, streets, tourist spots, and outdoor fitness areas. Illumination sources include street lights, lamps, flashlights, and candles.

**EgoNight-Oxford.** Oxford Day–Night Wang et al. (2025b) is a notable exception that also includes egocentric videos captured under both daytime and nighttime conditions. Although it was originally designed for 3D vision tasks such as novel view synthesis, it offers illumination variations across five representative locations in Oxford. However, the day and night videos are not spatially or temporally aligned. To enrich EgoNight with more realistic nighttime content, particularly for urban outdoor

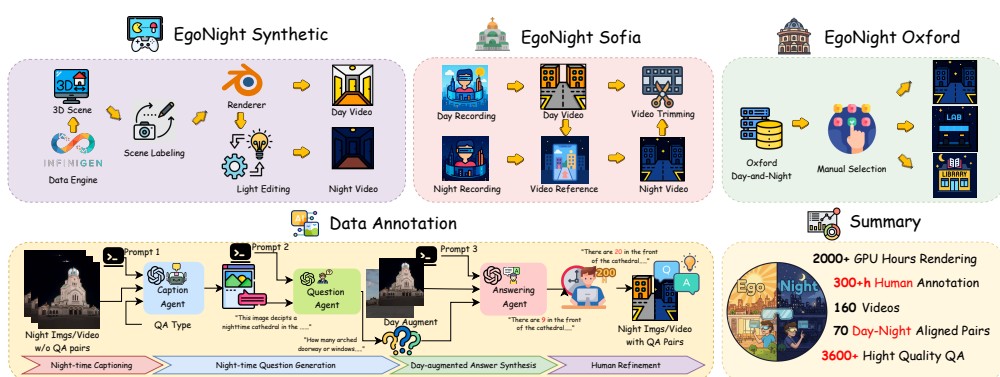

Figure 2: **EgoNight construction and EgoNight-VQA annotation.** EgoNight integrates EgoNight-Synthetic, EgoNight-Sofia, and EgoNight-Oxford sources. Annotation is achieved via a novel three-stage day-augmented Auto QA generation pipeline with 300+ hours of human refinement, resulting in over 3600 high-quality QA pairs.

scenes, we manually select 20 nighttime segments to form EgoNight-Oxford, based on two criteria: (i) minimal overlap in trajectories and locations, and (ii) genuinely low-light conditions. These segments serve as a complementary testbed for evaluating model generalization under illumination changes when paired alignment is unavailable.

For both EgoNight-Sofia and EgoNight-Oxford, human annotators categorize each video into easy, medium, or hard levels. Together, these sources provide EgoNight with a balanced combination of precise alignment, natural dynamics, and broad real-world diversity.

### 3.2 EGONIGHT-VQA BENCHMARK RECONSTRUCTION

**QA Task Taxonomy.** To thoroughly assess models from multiple perspectives, we define a diverse taxonomy of 12 QA tasks. Some of these categories are less studied or newly proposed in EgoNight-VQA, including scene sequence and navigation (which require not only visual perception but also memory and spatial reasoning), illumination recognition and illumination change (designed to test models' understanding of lighting concepts), and non–common-sense reasoning (e.g., detecting abnormal cases such as a door inserted into a wall in the synthetic data). More detailed explanations of QA types can be found in the Appendix Sec. A.4.1. We further organize these categories into *paired* and *unpaired* QA types, depending on whether the same questions can be consistently applied across day–night counterparts: 1) **Paired QA Types.** These cover contexts that remain unchanged across day and night, allowing the same QA pairs to be used for both videos and thus providing a clean testbed for measuring performance gaps. Specifically, we include: ① *object recognition*, ② *text recognition*, ③ *spatial reasoning*, ④ *scene sequence*, ⑤ *navigation*, ⑥ *counting of static*, ⑦ *action recognition*, and ⑧ *non–common-sense reasoning*. 2) **Unpaired QA Types.** These include categories that are impractical to pair across day and night, or are only meaningful in the nighttime condition. We consider: ① *lighting recognition*, ② *lighting dynamic*, ③ *dynamic detection*, and ④ *counting of dynamic*. We control QA clip duration by task type. For static or spatial tasks (e.g., object recognition, lighting recognition), we use short clips of 3 seconds to minimize redundancy; For dynamic or temporal tasks (e.g., action recognition, navigation), the entire video is used to capture the complete context. Following recent works Plizzari et al. (2025); Xiao et al. (2025), we adopt the *open-ended QA* setting over the closed-form multiple-choice format, as it better reflects natural human–AI interactions. A detailed summary of each QA type, including whether it is paired or unpaired, clip duration, and example questions, is provided in Fig. 3. This taxonomy makes EgoNight-VQA not only diverse and well-structured but also novel, introducing illumination reasoning and other challenges uniquely tied to nighttime egocentric vision.

**Day-Augmented Auto QA Generation.** Constructing QA pairs for nighttime videos is challenging because low visibility makes direct annotation time-consuming and error-prone. To address this, we design a three-stage *day-augmented auto QA generation* pipeline, shown in Fig. 2, that uses aligned daytime videos as a strong prior for annotating nighttime clips. The pipeline is tailored to each QA type and consists of: 1) **Nighttime captioning**, where GPT-4.1 generates detailed captions focused on the target QA type, such as object attributes or text/logos; 2) **Nighttime question generation**, where the caption and nighttime clip are used to produce diverse, visually grounded question

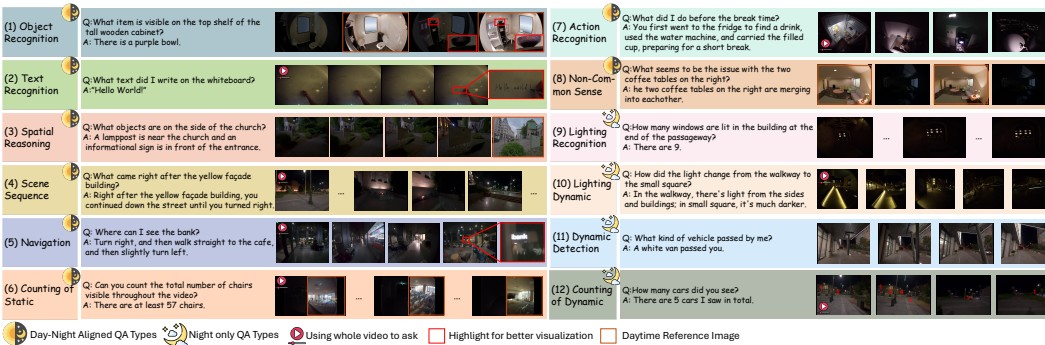

Figure 3: **QA types with examples.** The first eight are *paired* types, where the same question–answer applies to both day and night clips; the last four are *unpaired*, evaluated only at night. QA Types have various durations, with static or spatial tasks (e.g., 1 and 3) using short clips, while dynamic or temporal tasks (e.g., 4 and 5) use full videos.

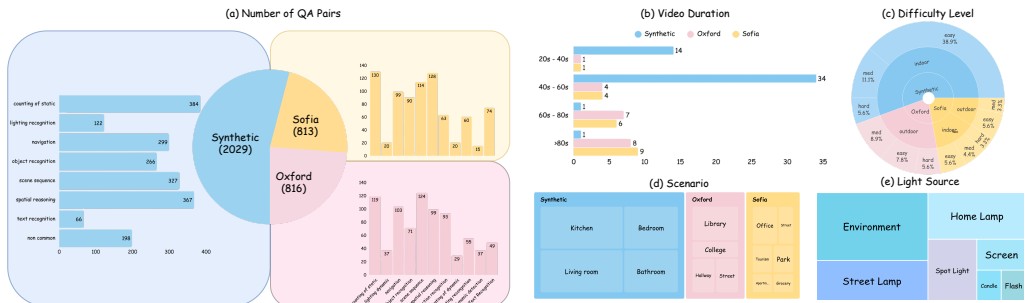

Figure 4: **Statistics of EgoNight-VQA benchmark.** (a) Distribution of QA pairs across QA types and sources. (b) Video duration distribution. (c) Task difficulty levels cross scenarios. (d) Scenario coverage. (e) Illumination coverage.

candidates; and 3) **Day-augmented pseudo-answer synthesis**, where answers for paired QA types are generated using the aligned daytime clip, while unpaired data such as EgoNight-Oxford relies directly on the nighttime clip. Empirically, both QA-type-specific prompting and daytime augmentation improve the quality and reliability of the generated QA pairs. More examples are provided in Appendix A.4.2 and Appendix A.7.1.

**Human Annotator Refinement.** Finally, human annotators refine QA pairs via three operations: i) **delete**, when QA pairs are meaningless, vague, duplicated, or inconsistent across day–night counterparts (for paired QA types); ii) **modify**, when the question is valid but the answer is wrong (or vice versa), or to resolve ambiguity; iii) **add**, when many pairs are removed or when important, challenging questions, especially about dynamic concepts, are missing. After the first labeling round, we performed a random double-check to refine low-quality annotations. Thus, although our pipeline combines model generation with human refinement, *every QA pair (3,658 in total) is manually verified at least once*. In total, ∼200 hours of human effort were invested, ensuring the quality and reliability of EgoNight-VQA.

**Dataset Statistics.** EgoNight-VQA comprises 3,658 high-quality, fully human-verified QA pairs across 12 task types, sourced from EgoNight-Synthetic, EgoNight-Sofia, and EgoNight-Oxford, with an average of 40 pairs per video. Detailed statistics on QA distribution, video durations, task difficulties, scenarios, and illumination are shown in Fig. 4. The number of videos across the three subsets—Synthetic (50), Sofia (20), and Oxford (20)—is proportionally reflected in the VQA annotations (2029 : 813 : 816). This results in an approximately 1:1 balance between synthetic and real (Sofia + Oxford) VQA samples, ensuring that our benchmark is not dominated by synthetic content. We provide more comparison of Egocentric VQA datasets in Appendix A.3. Overall, EgoNight-VQA provides a diverse and comprehensive benchmark for evaluating egocentric vision models under nighttime conditions.

## 3.3 BENCHMARKS BEYOND EGOCENTRIC VQA

**Day–Night Correspondence Retrieval.** Beyond VQA, we introduce *day–night correspondence retrieval* to evaluate whether models can match egocentric visual content across illumination conditions. This benchmark includes two subtasks: spatial retrieval and temporal localization. **Spatial retrieval** evaluates scene-level recognition across clips. Given a query clip and a set of $N$ candidate clips of equal duration $s$, the model must retrieve the candidate depicting the same scene. This tests whether the model can capture spatial relations in egocentric videos, such as distinguishing a bedroom from a bathroom or from another bedroom. We construct 1000 randomly generated meta-tasks, where each task samples a query clip and a candidate set containing its temporally aligned counterpart, with a temporal shift for added difficulty, together with $N - 1$ negatives from other scenes. Performance is measured by Top-1 accuracy. We set $N = 10$, $s = 10$ seconds, and use a temporal shift of [10, 20] frames. We evaluate both *Day → Day* and *Day → Night* settings. **Temporal localization** evaluates fine-grained temporal alignment across videos. Given a query clip of duration $s$, the model must localize it within the corresponding full video by predicting its start and end timestamps $(t_i, t_j)$, e.g., grounding "The door is being locked" to 10–20s. We construct 1000 meta-tasks by randomly sampling clips from randomly selected parent videos. Following temporal grounding literature Xin et al. (2024), we use mean Intersection-over-Union (mIoU) between the predicted interval $(t_i, t_j)$ and the ground-truth interval $(t_i^*, t_j^*)$ as the evaluation metric. Consistent with spatial retrieval, we set $s = 10$ seconds and evaluate both *Day → Day* and *Night → Day* settings.

**Egocentric Depth Estimation at Night.** Depth estimation is a fundamental component of computer vision. On the one hand, extensive research Yang et al. (2024a;b); Wang et al. (2025a) has focused on depth estimation in non-egocentric settings (typically not with fisheye cameras), while egocentric depth estimation remains largely underexplored, especially under nighttime conditions. On the other hand, recent works Chen et al. (2024a); Liu et al. (2025) suggest that incorporating depth can enhance models' spatial reasoning abilities. These two observations motivate us to construct an auxiliary benchmark for *egocentric depth estimation at night*. Specifically, we use EgoNight-Synthetic as the testbed, where ground-truth depth maps are provided by the rendering engine. Thanks to the day–night aligned design, we can quantitatively evaluate models under both controlled daytime and nighttime conditions. For evaluation, we adopt standard depth estimation metrics, including absolute relative error (AbsRel), $\delta_1(1.25)$, $\delta_2(1.25^2)$, and $\delta_3(1.25^3)$, where $\delta_k$ measures the percentage of predicted pixels whose relative error is within a threshold of $1.25^k$.

## 4 EXPERIMENTS

### 4.1 EVALUATED MLLMS & METRICS

We evaluate a broad set of state-of-the-art MLLMs on the proposed benchmarks. i) For **EgoNight-VQA**, we include two closed-source commercial models, GPT-4.1 Achiam et al. (2023) and Gemini 2.5 Pro Comanici et al. (2025); eight open-source models, Qwen2.5-VL (3B, 7B, 72B) Bai et al. (2023), VideoLLaMA3 (7B) Zhang et al. (2023b), InternVL3 (8B) Chen et al. (2024b), GLM-4.1V (9B-Base) Hong et al. (2025), and LLaVA-NeXT-Video (7B) Li et al. (2024); as well as EgoGPT Yang et al. (2025), one of the few open-source egocentric models tailored for open-ended QA. Following prior work Plizzari et al. (2025); Fan (2019), we adopt an *LLM-as-a-Judge* strategy to assess semantic consistency between predictions and ground truth, and report average accuracy across the test sets. ii)We provide further in-depth analysis on synthetic data quality, and potential model improvements. iii) For **day–night correspondence retrieval**, we benchmark feaFVisture-based retrieval methods, DINOv2 Oquab et al. (2023) and Perception Encoder (Percep. Enc.) Bolya et al. (2025), alongside MLLM-based methods, GPT-4.1 and InternVL3 (8B). As described in Sec. 3.3, Top-1 accuracy (Acc-R@1, %) and mIoU (%) are used for evaluating the spatial and temporal subtasks, respectively. iv) For **egocentric depth estimation**, we test a general monocular depth model (Depth Anything Yang et al. (2024a;b)), a 3D reconstruction-based method (VGGTStream Zhuo et al. (2025); Wang et al. (2025a)), and two egocentric fisheye-specific models (DAC Guo et al. (2025) and UniK3D Piccinelli et al. (2025)). For Depth Anything and VGGTStream, input fisheye RGB frames and depth maps are undistorted prior to inference for fair comparison. Additional implementation details (e.g., fps for frame extraction, prompts, and model settings) and discussion about LLM-as-a-Judge strategy are provided in the Appendix Sec. A.5.

| Models | EgoNight-Synthetic | | | EgoNight-Sofia | | | EgoNight-Oxford | | | Avg. |
|---|---|---|---|---|---|---|---|---|---|---|
| | Easy | Medium | Hard | Easy | Medium | Hard | Easy | Medium | Hard | - |
| *Closed-Source MLLMs* | | | | | | | | | | |
| GPT-4.1 | 29.30 | **26.87** | **18.87** | 32.04 | **29.35** | **31.69** | **39.72** | **37.13** | **40.72** | **30.93** |
| Gemini 2.5 Pro | **31.05** | 24.81 | 16.51 | **38.24** | 26.81 | 28.87 | 36.75 | 36.81 | 27.88 | 30.60 |
| *Open-source MLLMs* | | | | | | | | | | |
| InternVL3-8B | 20.21 | 15.50 | 16.98 | 24.03 | 21.74 | 20.42 | 22.90 | 20.85 | 16.36 | 20.06 |
| Qwen2.5-VL-72B | 18.39 | 15.25 | 12.26 | 24.03 | 17.03 | 20.42 | 24.81 | 22.80 | 16.36 | 18.99 |
| Qwen2.5-VL-7B | 13.01 | 13.95 | 13.68 | 15.44 | 12.68 | 12.68 | 13.74 | 13.36 | 12.73 | 13.44 |
| Qwen2.5-VL-3B | 14.69 | 10.34 | 7.08 | 15.50 | 13.04 | 12.68 | 17.18 | 11.40 | 12.12 | 13.41 |
| GLM-4.1V-9B-Base | 19.09 | 13.70 | 15.57 | 18.60 | 18.48 | 16.20 | 17.15 | 22.15 | 18.79 | 18.20 |
| VideoLLaMA3-7B | 16.85 | 13.44 | 14.62 | 11.11 | 10.87 | 9.15 | 12.26 | 10.46 | 9.15 | 13.64 |
| LLaVA-NeXT-Video-7B | 6.36 | 11.37 | 1.89 | 13.95 | 9.78 | 14.79 | 3.05 | 2.61 | 3.03 | 7.28 |
| *Egocentric MLLMs* | | | | | | | | | | |
| EgoGPT | 15.79 | 13.55 | 12.04 | 12.41 | 12.13 | 10.36 | 12.37 | 13.58 | 13.68 | 14.29 |

Table 1: **Comparison results on EgoNight-VQA.** Accuracies (%) of OpenQA results across three datasets and three difficulty levels. We compare closed-source models, open-source models, and egocentric-specific models.

## 4.2 RESULTS ON EGONIGHT-VQA

The main results of all MLLMs are shown in Tab. 1. In addition, we provide per-QA performance comparisons between day (striped bars) and night (solid bars) for paired QA types (Fig. 5(a)) and report nighttime performance across all QA types (Fig. 5(b)), based on averages across all models. Note that non–common case detection is available only in EgoNight-Synthetic, while dynamic events and actions are included only in the real-world data. From the results in Tab. 1, we observe that almost all MLLMs struggle on our benchmark, and the wide performance spread also confirms that our dataset is sufficiently challenging and effective for distinguishing model capabilities. Fig. 5(a) further highlights the performance gap, showing declines of 32.8% and 25.0% on EgoNight-Synthetic and EgoNight-Sofia, respectively. Together, these results underscore the substantial challenges posed by our benchmark, exposing the limitations of current MLLMs under nighttime scenarios and highlighting the need for more illumination-robust models. Beyond the overall trends, we note three additional insights from Tab. 1: i) Closed-source models perform best. Within open-source models, Qwen2.5-VL generally improves with scale, yet InternVL outperforms the larger Qwen2.5-VL (72B), suggesting that size alone is insufficient. The relatively low results of EgoGPT further emphasize the need for more robust egocentric models. ii) EgoNight-Oxford achieves the highest scores, but its illumination conditions are more challenging than those in EgoNight-Synthetic and EgoNight-Sofia (Sec. A.4.2, Appendix). This indicates that without paired day videos and our day-augmented auto-labeling strategy, even human annotators face difficulties generating challenging QA pairs, underscoring the practical value of our dataset design; iii) Overall, performance declines across task levels (easy, medium, hard), validating the diversity and difficulty of our benchmark. From the per-QA results in Fig. 5(a) and Fig. 5(b), we further observe three key trends: i) Models perform better on perception-oriented tasks (e.g., object recognition, text recognition, scene sequence) than reasoning-oriented tasks (e.g., navigation, counting, non-common-sense reasoning cases) under daytime conditions. However, at night, perception tasks suffer larger performance drops, indicating their higher sensitivity to illumination, whereas reasoning tasks, though harder overall, are relatively less affected since they rely more on temporal and contextual cues. ii) MLLMs achieve substantially lower accuracy on our newly proposed tasks, such as lighting recognition, lighting dynamics, scene sequence, dynamic detection, navigation, and non-common-sense reasoning, suggesting that existing MLLMs generalize poorly to novel tasks compared with well-studied ones like object recognition. iii) Each dataset in Fig. 5(b) emphasizes distinct aspects of nighttime challenges, together providing complementary perspectives that ensure EgoNight spans a balanced range of perception–reasoning difficulties under low-light conditions.

## 4.3 MORE IN-DEPTH ANALYSIS

**Quality of EgoNight-Synthetic.** More examples of synthetic visualizations are provided in Appendix A.4.2 and A.1. To quantitatively evaluate how well the synthetic data reflects real-world performance, we benchmark a diverse set of multimodal large language models (MLLMs) on three

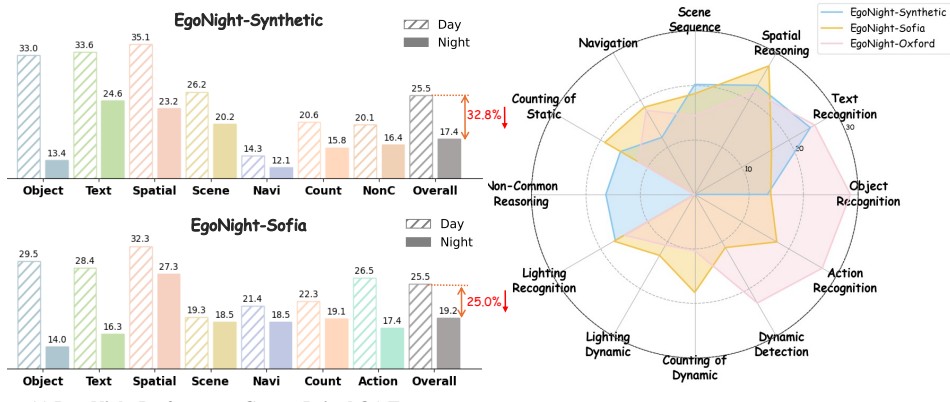

Figure 5: **Performance analysis of MLLMs on EgoNight-VQA.** (a) Day–night performance gap across paired QA types, showing consistent degradation at night. (b) Nighttime performance across all 12 QA types. NonC means non–common-sense reasoning.

subsets: Synthetic, Sofia (real), and Oxford (real). For each model, we compute its overall VQA accuracy on each subset and then calculate the Pearson correlation between the average per-model accuracies on synthetic and real-world data (Appendix A.6). We observe strong correlations between synthetic and Sofia ($0.9359$, p-value $6.847 \times 10^{-5}$), and between synthetic and Oxford ($0.8588$, p-value $1.462 \times 10^{-3}$). These results indicate that models that perform well on synthetic data also tend to perform well on real nighttime data, suggesting that the synthetic set preserves the relative difficulty and ranking of models. To further examine transferability, we fine-tune Qwen2.5-VL-7B using supervised fine-tuning (SFT) on the synthetic training split only, while keeping the real-world data unseen during training. We then evaluate the fine-tuned model directly on the real dataset. The accuracy improves from $14.83\%$ (zero-shot baseline) to $20.57\%$, demonstrating that supervision from synthetic nighttime scenes generalizes to real-world nighttime scenarios.

**Impact of Fine-Tuning.** To study model adaptation to nighttime egocentric QA, we split EgoNight into $70\%$ training and $30\%$ testing subsets, ensuring no video overlap between splits. We fine-tune Qwen2.5-VL-7B with supervised fine-tuning (SFT) under three configurations: i) full model fine-tuning (updating both vision encoder and LLM), ii) vision-encoder-only tuning (freezing the LLM), and iii) LLM-only tuning (freezing the vision encoder). All models are trained on the same training split and evaluated on the held-out test split. Tab. 2 reports the performance for synthetic and real (Sofia and Oxford) separately. It shows consistent performance gains across all configurations compared to the zero-shot baseline. Updating either the vision encoder or the LLM independently improves accuracy, while full fine-tuning achieves the best performance, indicating that both improved visual representations and stronger vision-language alignment contribute to adaptation under low-light conditions.

| Model | Synthetic | Real |
|---|---|---|
| Qwen7B (Base) | 23.23 | 16.40 |
| Enc. Only | 29.74 | 20.92 |
| LLM. Only | 35.50 | 22.26 |
| Full | 36.25 | 25.61 |

Table 2: Fine-tuning performance comparison across datasets.

| Task | Qwen7B | Enc. Only | LLM. Only |
|---|---|---|---|
| Object | 8.435 | 34.718 | 35.855 |
| Text | 18.440 | 49.890 | 50.988 |
| Navigation | 17.870 | 19.495 | 19.918 |
| Counting | 16.558 | 16.945 | 24.275 |

Table 3: Fine-tuning performance comparison across tasks.

**Perception vs. Reasoning under Fine-Tuning.** To better understand which capabilities benefit most from adaptation, we group tasks into perception-oriented (Object Recognition, Text Recognition) and reasoning-oriented (Navigation, Counting) categories and report their accuracies separately in Tab. 3. Perception-oriented tasks show larger absolute improvements after fine-tuning, suggesting that low-level visual adaptation plays a significant role in nighttime scenarios. Vision-encoder-only tuning primarily boosts perception tasks, whereas LLM-only tuning improves both perception and reasoning tasks to a certain extent. However, reasoning-oriented tasks exhibit comparatively smaller

| Models | Spatial Retrieval (Acc - R@1 % ↑) | | | | Temporal Localization (mIoU % ↑) | | | |
|---|---|---|---|---|---|---|---|---|
| | EgoNight-Synthetic | | EgoNight-Sofia | | EgoNight-Synthetic | | EgoNight-Sofia | |
| | Day→Day | Night→Day | Day→Day | Night→Day | Day→Day | Night→Day | Day→Day | Night→Day |
| DINOv2 | 45.7 | 28.7 | 84.5 | 74.5 | - | 33.7 | - | 33.1 |
| Percep. Enc. | 65.4 | 41.6 | 89.8 | 80.9 | - | 32.9 | - | 33.4 |
| GPT-4.1 | 75.6 | 54.1 | 92.5 | 84.5 | 14.7 | 10.0 | 21.2 | 15.5 |
| InternVL3-8B | 39.4 | 27.7 | 73.9 | 56.3 | 10.2 | 9.9 | 12.5 | 13.3 |

Table 4: **Night-to-Day retrieval performance.** Each dataset is evaluated on both Day→Day and Night→Day settings.

| Method | Abs Rel ↓ | | $\delta_1$ (1.25) ↑ | | $\delta_2$ (1.25$^2$) ↑ | | $\delta_3$ (1.25$^3$) ↑ | |
|---|---|---|---|---|---|---|---|---|
| | Day | Night | Day | Night | Day | Night | Day | Night |
| Depth Anything (U) | 0.297 | 0.302 | 0.249 | 0.237 | 0.463 | 0.447 | 0.622 | 0.60 |
| VGGTStream (U) | 0.293 | 0.298 | 0.234 | 0.232 | 0.447 | 0.442 | 0.615 | 0.609 |
| DAC (F) | 0.245 | 0.292 | 0.255 | 0.216 | 0.495 | 0.425 | 0.684 | 0.602 |
| UniK3D (F) | 0.224 | 0.253 | 0.280 | 0.254 | 0.524 | 0.481 | 0.706 | 0.658 |

Table 5: Depth estimation results on EgoNight-Synthetic. **U**: undistorted input; **F**: fisheye input.

gains overall, indicating that high-level reasoning under degraded visual input remains challenging even after adaptation.

**Failure Cases.** Additional analysis of failure cases is provided in Appendix A.8.1.

### 4.4 RESULTS ON OTHER TASKS

**Night-to-day retrieval** The results of day–night retrieval are reported in Tab. 4. The gap between Night–Day and Day–Day shows that cross-illumination retrieval remains highly challenging compared with in-domain retrieval. For spatial retrieval, GPT-4.1 consistently outperforms other methods, achieving over $80\%$ accuracy. This suggests that Retrieval-Augmented Generation methods could further improve performance, as Fig. 5(a) already shows that daytime inputs significantly benefit the models. For temporal retrieval, however, GPT-4.1, despite its strong results on egocentric VQA (Tab. 1) and spatial retrieval, shows a substantial drop compared with feature-based methods (DINOv2 and Perception Encoder). A similar degradation is observed for InternVL3-8B. These findings suggest that while MLLMs excel at spatial semantic understanding, they struggle with temporal reasoning, such as timestamp prediction, which is critical for temporal localization. Further results on temporal limitations are provided in Appendix A.6.

**Depth estimation.** Results for depth estimation are reported in Tab. 5. The relatively low scores across all models highlight the difficulty of our EgoNight dataset, which combines egocentric motion, complex geometry, and extreme lighting variations. A clear gap between daytime and nighttime performance again underscores the challenges of low-light conditions. Among the methods, fisheye-based methods (DAC and UniK3D) outperform general depth estimators, suggesting the need for egocentric-specific algorithms. Additional qualitative results are provided in Sec. A.7.3.

## 5 CONCLUSION

In this work, we introduce EgoNight, the first benchmark suite for systematically evaluating egocentric MLLMs under challenging nighttime conditions. EgoNight combines synthetic and real-world videos with day–night alignment, enabling rigorous analysis of illumination effects. Based on this data, we construct EgoNight-VQA, covering 12 QA types with 3,658 human-verified QA pairs, along with two auxiliary benchmarks: day–night correspondence retrieval and nighttime egocentric depth estimation. Experiments show that state-of-the-art MLLMs suffer substantial performance drops at night compared to daytime, indicating that nighttime egocentric vision remains far from solved. We hope EgoNight will facilitate future research on illumination-robust egocentric perception and reasoning, and support the development of more reliable egocentric AI assistants.

ACKNOWLEDGMENTS

This research was partially funded by a collaboration project between VIVO and INSAIT on indoor scene understanding and editing. Research was also funded by the Ministry of Education and Science of Bulgaria (support for INSAIT, part of the Bulgarian National Roadmap for Research Infrastructure) . This project was supported with computational resources provided by Google Cloud Platform (GCP). We sincerely thank Dr. Zirui Wang, the author of the Oxford Day-and-Night dataset, for the helpful discussions and for kindly providing the source videos.

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
