## A APPENDIX

### A.1 MORE VIDEO SOURCE CONSTRUCTION DETAILS

**EgoNight-Synthetic Construction.** For EgoNight-Synthetic Construction, we first use the coarse progressive generation method with a fast solver in infinigen Raistrick et al. (2023) to generate 3D scenes in Blender format. Then, a human annotator will edit the scene in the following sequence:

- Explore and edit the scene to remove unreasonable cases and make the indoor scene as natural as possible.
- Add light source in the scene if the generated scene does not include enough illumination to create enough illumination gap between the day and night.
- Record camera trajectory by exploring the whole indoor scene.
- Change the camera model and resolution. Set rendering samples and frames. For all synthetic dataset, we use the Blender build-in Panoramic Fisheye Equisolid camera with Lens 10.5 and field of view 180°.
- Create night scene by modifying the light source, motion blur, and environment map.
- Render the day and night pair using Blender Iraci (2013).

During the dataset construction, we apply home light source during night for 30 scenes and spot light source for 20 scenes to simulate torch light in real life. To create different difficulty levels, we apply different rendering sample size (higher sample size gives lower noise in the final image), spot light size, and motion blur to part of the data as shown in Tab. 6. We also show different modality and difficulty level in Fig. 6.

| difficulty level | sample size | light condition | motion blur shutter |
|---|---|---|---|
| Easy | 4096 | 105°/ few light on | - |
| Medium | 512 | 40°-50°spot light / all light off | - |
| Hard | 512 | 40°-50°spot light / all light off | 1-2 |

Table 6: Difficulty level and corresponding rendering settings.

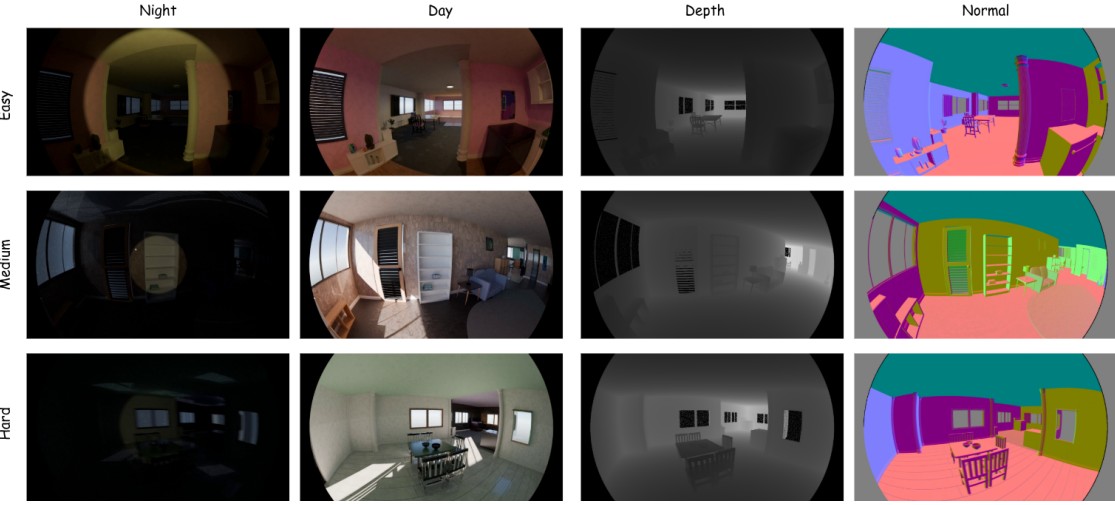

Figure 6: More examples and modalities of synthetic datasets.

**EgoNight-Sofia Construction**. In total, four participants were involved. The recording setup included three different GoPros, a head-mounted rig to fix the camera on the forehead and mimic human-eye perspective, several phones for live preview or daytime video guidance, and diverse lighting sources such as flashlights, spotlights, and candles. The process followed a video-guided recording strategy, as introduced in Sec. 3.1: the ego-wearer first recorded a daytime video while previewing the live feed on a phone, and for the nighttime counterpart replayed the daytime video on the phone as guidance to replicate the same setup, walking speed, viewpoints, and actions. Videos were collected across a wide range of environments, including indoor scenes (apartments, workplaces, grocery shops, building receptions) and outdoor scenarios (fitness areas, tourist landmarks, and street views). Post-trimming was applied to each day–night pair to further ensure alignment. On average, it took around 2-3 hours to produce one paired data.

**EgoNight-Oxford Construction.** We credit the contribution of Seeing in the Dark dataset Wang et al. (2025b), which provides multiple sequences of egocentric videos in the night in a various environment. We built our EgoNight-VQA dataset partially upon this work. Firstly, We enumerated all nighttime clips in Oxford Day–Night and performed a two-stage filtering. (1) Screening for uniqueness of place. We cross-checked scene metadata (route notes/time stamps) to avoid repeated paths within the same landmark. (2) Stratified diversity & quality sampling. Remaining clips were scored on a 1–5 rubric along axes designed for egocentric, low-light evaluation: illumination type (ambient only / mixed artificial / high-contrast point sources), illumination hardness (soft vs. specular/point), exposure stability (auto-gain pumping, blown highlights), scene dynamics (pedestrians/vehicles, occlusions), camera motion pattern (walk, run, head turns), and task context (navigation, road-crossing, object interaction, signage reading). The final set comprises 20 sequences that maximize lighting/task diversity under egocentric night settings while avoiding place overlap and task overlap.

In total, collecting the three video sources required over 100 hours of human effort.

## A.2 MORE BENCHMARK IMPLEMENTATION DETAILS

As described in Sec. 3.2, our auto-labeling pipeline is QA-type specific and involves three customized prompts: one for captioning, one for question generation, and one for answer synthesis. The detailed prompts are shown in Fig. 8.

### A.2.1 EGONIGHT-VQA HUMAN LABELING

We hired several participants to review and refine the QA pairs generated by our three-stage day-augmented auto-labeling pipeline, compensating them at a rate of €20 per video. Each participant was provided with a detailed labeling instruction document and an onboarding meeting to ensure the guidelines were clearly conveyed. A simplified version of the labeling tutorial is included as in Fig. 7.

---

**Annotation Tutorial (Simplified)**

**Read Me First:** Please follow the labeling pipeline carefully and complete each step as instructed. On average, annotating one video takes about 2 hours. Easier cases may take less time, but in general, each video should take no less than 1.5 hours to ensure high-quality annotations. (The first video may take longer, as you will need to familiarize yourself with the pipeline.) We will randomly check the labeled data afterward, and annotators will be required to refine their work if the quality does not meet expectations.

**Step 1: Preparation.** You are expected to first download the paired `day.mp4` and `night.mp4` videos (aligned in time, except for unpaired tasks), together with the QA text file (`.txt`), which contains candidate QAs grouped by QA type (e.g., `counting.txt`). Before starting annotation, you should carefully watch both the daytime and nighttime videos to fully understand the scenario and activities.

**Step 2: QA Verification and Refinement.** For each QA pair, you should apply one of three operations:

- **Delete**: You should remove QAs that are meaningless, vague, irrelevant, duplicated, or inconsistent between day–night pairs (for paired QA types).

- **Modify**: If the question is reasonable but the answer is incorrect (e.g., counting errors, wrong action duration), you should correct the answer. You may also rephrase the question to eliminate ambiguity (e.g., clarifying "left/right" as relative to the ego-wearer).

- **Add**: If too many pairs are deleted, or if you notice interesting and challenging cases missing, you need to add new QAs. This is especially important for low-frequency tasks, e.g., dynamic detection or counting of dynamics.

**Step 3: Special Cases.**

- For paired QA types (e.g., object recognition, spatial reasoning), you must ensure the same QAs apply to both day and night videos.

- For unpaired QA types (e.g., lighting recognition, dynamic detection), you only need to ensure correctness on the nighttime video.

- For dynamic events, you are expected to specify temporal spans, e.g., *Q: "Around which time does a red car pass by?" A: "At frames 4–6."*

**Step 4: Post-processing.** Once QAs were validated, the answer field should be renamed from `` ``answer'' `` to `` ``human_answer'' ``.

**Appendix: Paired & Non-Paired QA Types.** The same as described in the main file.

---

Figure 7: Simplified version of annotation tutorial.

| QA Type | Object Recongnition | Spatial Reasoning | Text Recongnition | Scene Sequence |
|---|---|---|---|---|
| Caption Prompt | | | | |
| Question Prompt | | | | |
| Answering Prompt | | | | |

| QA Type | Light Recognition | Light Change | Counting | Dynamic Counting |
|---|---|---|---|---|
| Caption Prompt | | | | |
| Question Prompt | | | | |
| Answering Prompt | | | | |

| QA Type | Non-Common | Navigation | Dynamic Recognition | Action |
|---|---|---|---|---|
| Caption Prompt | | | | |
| Question Prompt | | | | |
| Answering Prompt | | | | |

Figure 8: The prompts used during auto-labeling.

After the first round of labeling, we performed an additional quality check and summarized common issues for refinement. For example, the concept of "left" or "right" should always be defined relative to the ego-wearer, and questions should be phrased clearly to avoid ambiguity. Participants were then asked to address the identified issues. In the end, we ensured that every QA pair in the EgoNight-VQA dataset was verified by at least one human annotator. On average, above 200 hours of human annotation effort were spent for the labeling refinement. To further evaluate inter-

annotator statistics, we select randomly VQA pairs from 20 videos out of 90, and add four human annotators without GPT annotation. The average pairwise cosine similarity normalized to $(0, 1)$ based on language feature extracted by BLIP-2 Li et al. (2023b) is 0.8458. This shows that human annotators are in general consistent in interpreting scenes. And answers are semantically aligned, even when worded differently.

## A.3 COMPARISONS WITH PRIOR EGOCENTRIC VQA BENCHMARKS.

In Tab. 7, we compare our EgoNight-VQA with prior egocentric VQA benchmarks, including EgoVQA Fan (2019), EgoTaskQA Jia et al. (2022), EgoSchema Mangalam et al. (2023), EgoThink Cheng et al. (2024), EgoTempo Plizzari et al. (2025), EgoCross Li et al. (2025b), EgoMemoria Ye et al. (2024), HourVideo Chandrasegaran et al. (2024), and EgoLifeQA Yang et al. (2025), listing their lighting conditions (mainly daytime or nighttime), video duration length, the number of testing QA examples, number of QA type categories, if temporal-oriented tasks are included or emphasized, and the evaluation metric.

We highlight that EgoNight-VQA is the first to explore nighttime egocentric VQA with aligned day–night video pairs.

| Dataset | Lighting | Video Length | # Test | # Categories | Temporal | Metric Type |
|---------|----------|--------------|--------|--------------|----------|-------------|
| EgoVQA | ☼ | (25s, 100s) | 250 | 3 | ✗ | OpenQA |
| EgoTaskQA | ☼ | 25s | 8k | 4 | ✗ | OpenQA |
| EgoSchema | ☼ | 180s | 500 | - | ✗ | CloseQA |
| EgoThink | ☼ | - | 750 | 12 | ✗ | OpenQA |
| EgoTempo | ☼ | 45s | 500 | 10 | ✓ | OpenQA |
| EgoCross | ☼ | 23s | 957 | 15 | ✓ | CloseQA & OpenQA |
| EgoBlind | ☼ | (0s, 120s) | 5311 | 6 | ✓ | OpenQA |
| EgoMemoria | ☼ | (30s, 1h) | 7026 | - | ✓ | CloseQA |
| HourVideo | ☼ | (20min, 120min) | 12976 | - | ✓ | CloseQA |
| EgoLifeQA | mostly ☼ | 44.3 h | 6000 | 5 | ✓ | CloseQA |
| **EgoNight-VQA** | Aligned ☼ & ☾ | (24s, 214s) | 3658 | 12 | ✓ | OpenQA |

Table 7: Comparison between EgoNight-VQA and prior egocentric VQA benchmarks. ☼ means dayytime, while ☾ indicates nighttime.

## A.4 MORE EGONIGHT-VQA EXPLANATIONS AND EXAMPLES

### A.4.1 QA TYPE DEFINATION

We present the 12 QA types with their detailed definitions in Tab. 8.

Table 8: **Detailed descriptions of the 12 QA types in EgoNight-VQA.** Paired QA types share the same QAs across day–night counterparts, while unpaired QA types are evaluated only on nighttime videos.

| QA Type | Attribute | Description |
|---------|-----------|-------------|
| Object Recognition | Paired | Identify and recognize specific objects in the scene (e.g., "What is on the table?"). |
| Text Recognition | Paired | Read and interpret visible text or logos (e.g., "What does the sign say?"). |
| Spatial Reasoning | Paired | Understand spatial relations between objects (e.g., "What is left of the chair?"). |
| Scene Sequence | Paired | Recall the temporal order of visited scenes (e.g., "Which room did I enter after the kitchen?"). |
| Navigation | Paired | Working as an navigation assistant after watched the whole video (e.g., "How can I reach place B from place A?"). |
| Counting of Statics | Paired | Count static objects visible in the scene (e.g., "How many chairs are in the room?"). |
| Action Recognition | Paired | Identify human actions or interactions (e.g., "What action is being performed?"). |
| Non-Common-Sense Reasoning | Paired | Judge unusual or physically implausible cases, for synthetic videos. (e.g., "Is the door embedded inside the wall?"). |
| Lighting Recognition | Unpaired | Recognize the illumination source, also include counting. (e.g., "How many light sources are in the room?"). |
| Lighting Change | Unpaired | Detect changes in lighting conditions (e.g., "Did the light turn off during the clip?"). |
| Dynamic Detection | Unpaired | Detect dynamic moving objects (e.g., "Is a car/person moving across the scene?"). |
| Counting of Dynamics | Unpaired | Count the number of dynamic objects or events (e.g., "How many people walked by?"). |

### A.4.2 QA EXAMPLES

We show more QA examples of EgoNight-Synthetic in Fig. 9, EgoNight-Sofia in Fig. 10, and EgoNight-Oxford in Fig. 11. For those paired QA types, we show both day and night frames, while for those unpaired QA types, we demonstrate nighttime frames only. Three frames are shown if the QA is spatial or static related, while more frames are given if the QA is temporal or more dynamic related.

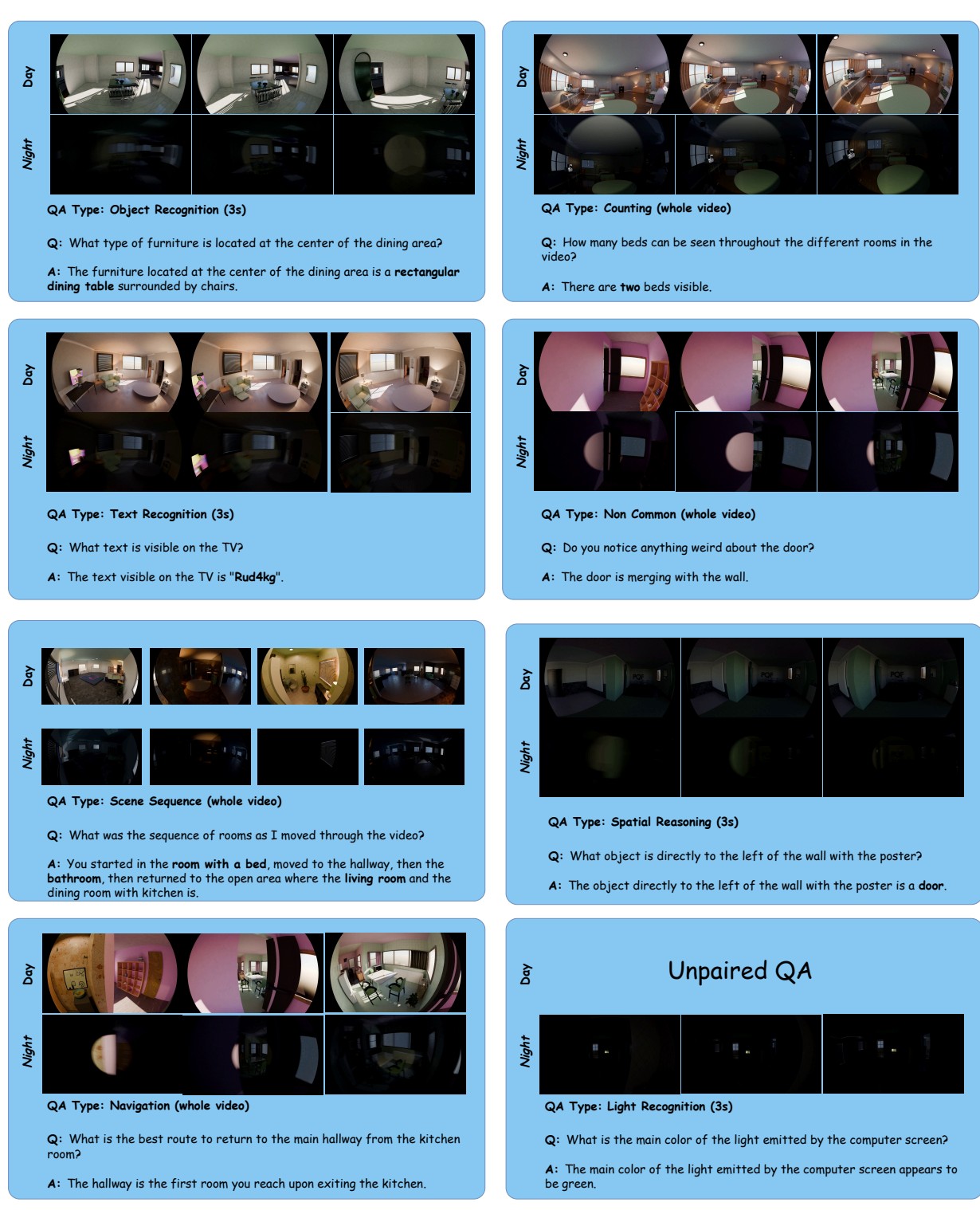

Figure 9: More QA examples from EgoNight-Synthetic dataset.

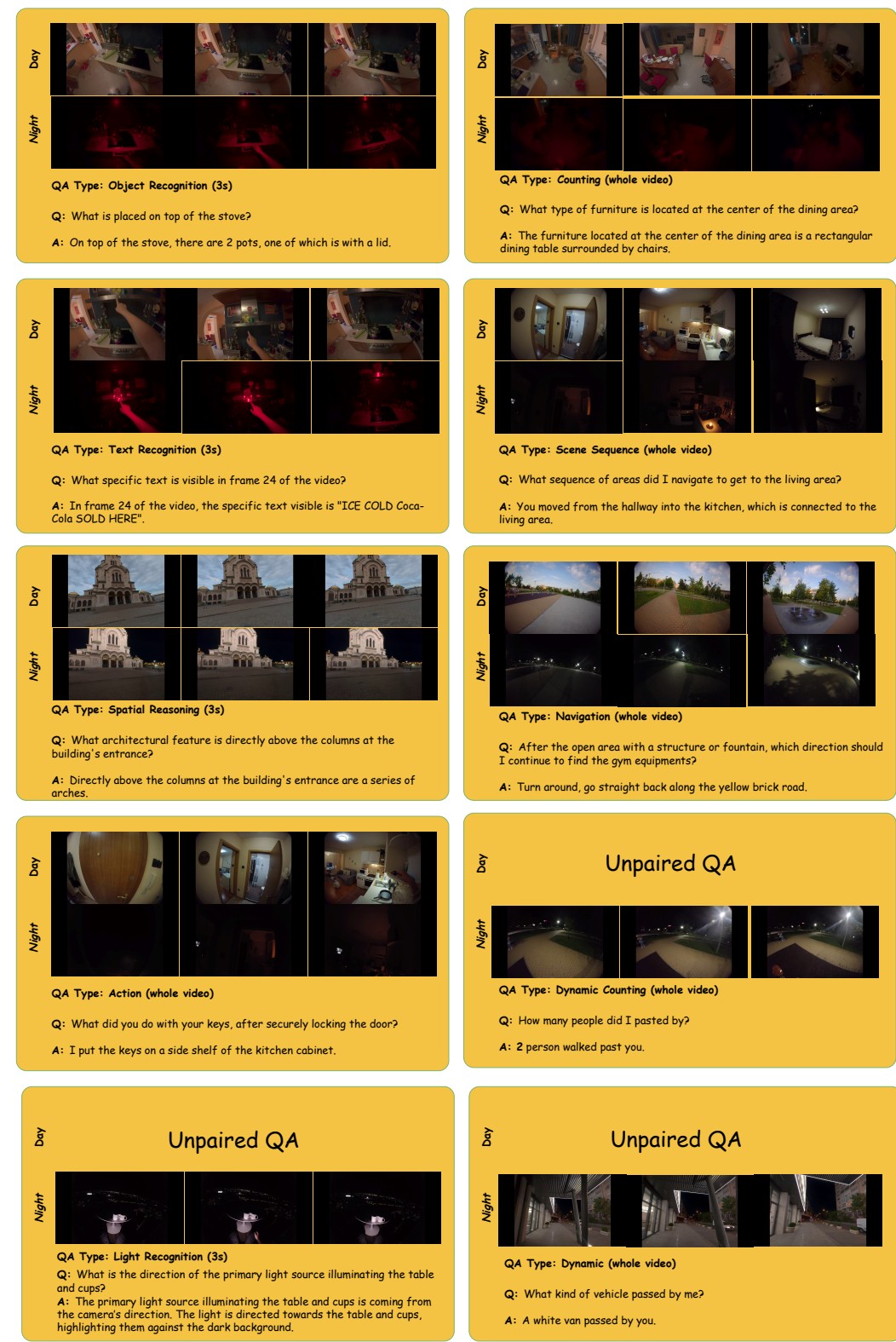

Figure 10: More QA examples from EgoNight-Sofia dataset.

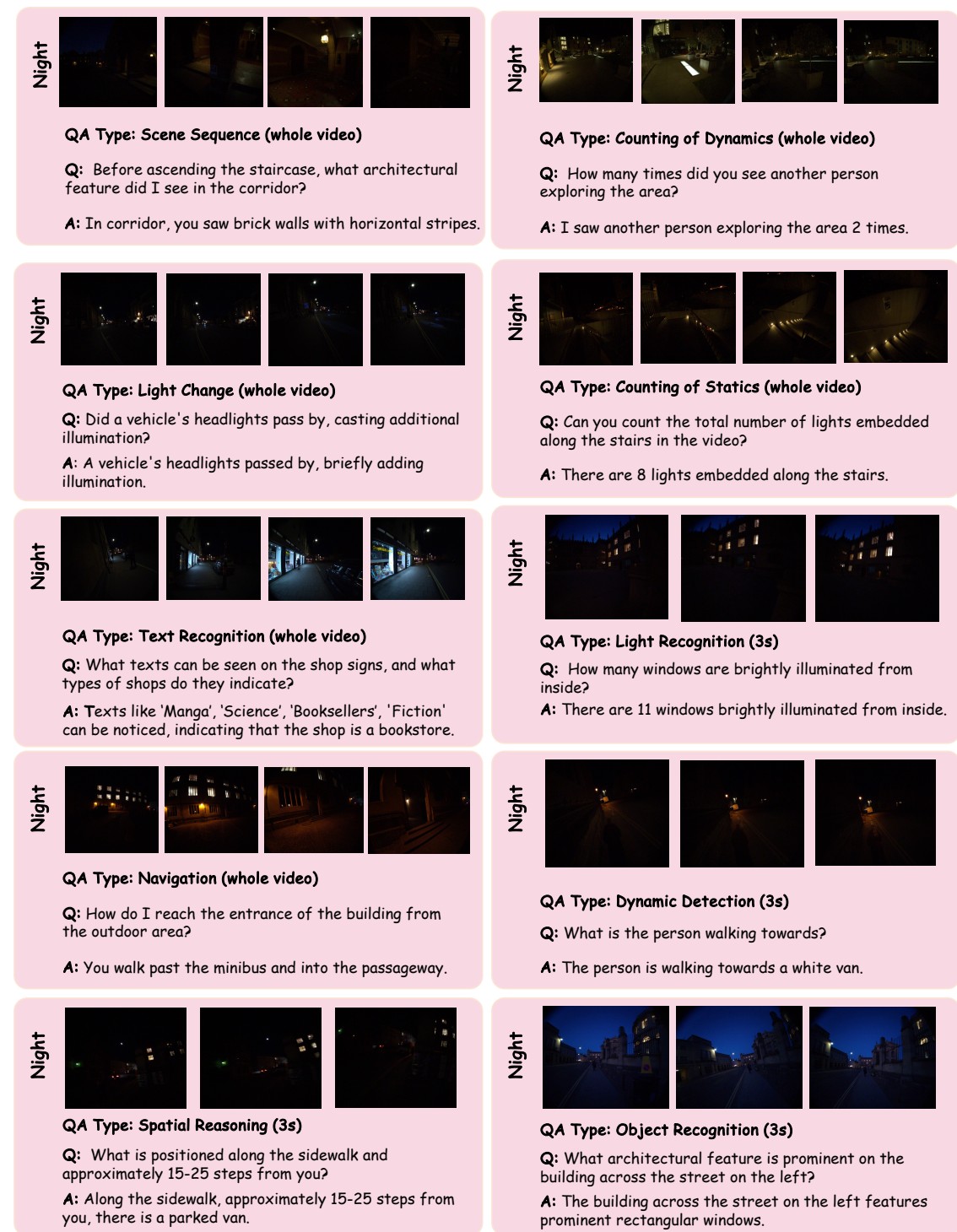

Figure 11: More QA examples from EgoNight-Oxford dataset.

## A.5 MORE EXPERIMENTS SETUPS

### A.5.1 SETUPS FOR EGONIGHT-VQA EXPERIMENTS.

In this section, we describe the model setup, how GPT is used as the judge, the prompting strategy, the GPU resources, and the approximate runtime for each dataset. For the closed-source model, we directly use the API call.

| Model | Inference Speed (min) |
|---|---|
| GPT-4.1 | <5 |
| Gemini 2.5 Pro | <5 |
| InternVL3-8B | <5 |
| Qwen2.5-VL-72B | 25 |
| Qwen2.5-VL-7B | <5 |
| Qwen2.5-VL-3B | <5 |
| GLM-4.1V-9B-Base | <5 |
| VideoLLaMA3-7B | <5 |
| LLaVA-NeXT-Video-7B | 50 |
| EgoGPT | <5 |

Table 9: Inference speed for different models (per Video).

For open source models, we use LLama-Factory Zheng et al. (2024) except VideoLLama3 Zhang et al. (2023b) and EgoGPT Yang et al. (2025). We use NVIDIA A6000 GPUs for all the model, except for Qwen2.5-VL-72B, we use 2 NVIDIA H200 GPUs for larger GPU memory. The inference speed for each model is shown in Tab. 9. Video frames are sampled at 2 fps for EgoNight-Synthetic, and 1 fps for EgoNight-Sofia and EgoNight-Oxford, without imposing a maximum frame limit. To further ensure fairness and consistency, the exact prompts used for each task are provided below.

**Model Evaluation Prompt.** For evaluating the language model, we use the following prompt:

*Please carefully read the question, use the visual cues in the {video} to answer the question: {question}.*

*The original FPS of the video is {original_video_fps}. This image set is obtained by sampling at {sampling} fps.*

*Do not include any other content. You need to answer the question in any case and not demand additional context information. Note: All the actions mentioned refer to the person who recorded the video.*

**Evaluation Protocol.** Since the questions and answers are open-ended, we utilize GPT-4.1 Achiam et al. (2023) as a judge. Here is the prompt for evaluating the score given the model prediction, the ground truth answer, and the corresponding question:

```
role: system,
content: You are an intelligent chatbot designed for evaluating the correctness of AI assistant predictions for question−answer
    pairs.
Your task is to compare the predicted answer with the ground−truth answer and determine if the predicted answer is correct or
    not. Here's how you can accomplish the task:
INSTRUCTIONS:
1. Focus on the correctness and accuracy of the predicted answer with the ground−truth.
2. Consider uncertain predictions, such as 'it is impossible to answer the question from the video', as incorrect, unless the ground
    truth answer also says that.
role: user,
content: Please evaluate the following video−based question−answer pair:
Question: {question}
Ground truth correct Answer: {answer}
Predicted Answer: {predicted_answer}
Provide your evaluation as a correct/incorrect prediction along with the score where the score is an integer value between 0 (fully
    wrong) and 5 (fully correct). The middle score provides the percentage of correctness. For question that counting the
    number of objects, if the predicted answer fells in the range of the ground truth answer, it should be considered as correct.
Please generate the response in the form of a Python dictionary string with keys 'pred', 'score' and 'reason', where value of 'pred
    ' is a string of 'correct' or 'incorrect',
value of 'score' is in INTEGER, not STRING and value of 'reason' should provide the reason behind the decision."
```

To further validate the LLM-as-a-Judge strategy, we divide all annotations into two groups: (a) answers verified but not modified by humans (preserving the GPT style) and (b) answers modified or created by humans. The ratio is approximately $4 : 6$, indicating a high human modification rate. When evaluating the accuracy of GPT-4.1 separately in the two subsets, we obtain $26.73\%$ for group (a) and $27.87\%$ for group (b). The similar scores show that GPT-as-Judge does not prefer GPT-generated answers over human-authored ones. Further more, We randomly sampled 300 QA pairs

(questions, ground truth, model answers, and the corresponding LLM-assigned scores) and asked human evaluators to judge whether each score from the LLM was correct. This yielded an agreement rate of $95.67\%$, indicating strong alignment between human judgment and the LLM-as-a-Judge decisions, and thus demonstrating the reliability of the LLM-based evaluation.

### A.5.2 Setups for Day-Night Correspondence Retrieval.

In this section, we describe the setup of the model, method, e.g. how feature-based retrieval for vision encoders, how prompt VLMs, metric, result, GPU, cost time, and other details.

**i) Spatial Retrieval (Place Recognition).** For feature-based methods Oquab et al. (2023); Bolya et al. (2025), we calculate the CLS tokens $f^i$ of each frame within the video clip with the vision encoder, with $i$ the frame index in the clip. Then, the "best matching" strategy is implemented to calculate the similarity between the query clip $v_q$ and the database clip $v_d$. The best cosine similarity between the features of the query clip $f_q^i$ and the database clip $f_d^j$,

$$\sigma(v_q, v_d) = \max_{i \in [0, s-1], j \in [0, s-1]} cos(f_q^i, f_d^j). \tag{1}$$

The database video clips are ordered based on the similarity $\sigma$ and then the most similar clip is retrieved.

Similarly, for MLLM-based methods Achiam et al. (2023); Chen et al. (2024b), we ask the MLLM to assess the "pairwise" similarity between each query-database pair and order the database clips by similarity. The prompt to the MLLM is as follows:

---

*You are given two video clips from different scenes.*
*Your task is to evaluate how similar these two scenes are based on their spatial layout, furniture, objects, architectural features, and overall room structure.*

*CLIP STRUCTURE:*
*− Images 1−>(s−1) from Query Scene*
*− Images s−>{2s−1} from Database Scene*

*TASK:*
*Please carefully analyze and compare the spatial layout, furniture placement, objects, architectural features, and overall room structure between these two video clips.*

*IMPORTANT: Please respond with ONLY a single numerical similarity score between 0.0 and 1.0, where:*
*− 0.0 = Completely different scenes (different rooms/locations)*
*− 1.0 = Identical or nearly identical scenes (same room/location)*
*− Values in between represent varying degrees of similarity*

*Example responses: "0.85", "0.23", "0.67"*
*1.0 should be used when the two scenes are identical, so don't use 1.0 if the two scenes are not 100% identical.*
*Please provide only the numerical score without any additional text or explanation.*

---

It is noticeable that existing MLLMs have difficulty in processing long-horizon and multi-scene videos. We also conduct the "all-in-one-prompt" experiments by inputting all the images of the query clip and the database clips in one prompt and asking the MLLM to output the ordered database clips. The "all-in-one-prompt" strategy leads to largely degraded performance, as shown in Tab. 10.

| | Spatial Retrieval R@1 - Synthetic | |
| Prompt Strategy | Day → Day | Night → Day |
|---|---|---|
| Pairwise | 75.6 | 54.1 |
| All-in-one | 10.5 | 28.5 |

Table 10: Ablation on prompting for night-to-day spatial retrieval task.

**ii) Temporal Localization.** The mIoU metric is defined as:

$$\text{mIoU} = \frac{1}{M} \sum_{m=1}^{M} \frac{\left| [t_i, t_j] \cap [t_i^*, t_j^*] \right|}{\left| [t_i, t_j] \cup [t_i^*, t_j^*] \right|}, \tag{2}$$

where $M$ denotes the total number of meta-tasks (1000 in our setup). For feature-based temporal localization, we apply the "best-match" strategy similar as spatial localization, localizing the query clip to the frame stamp with the best clip-to-clip similarity:

$$i = \arg\max_i \sigma(v_q, v_d), v_d = v_D[i : i + s], \tag{3}$$

$v_D$ is the parent full video and the end frame will be $i + s - 1$. For MLLM-based method, we input the query clip and the parent full video in the prompt and ask the MLLM to output the start and end frame of the query within the full video. The prompt is as follows:

---

*You are given a query video clip and a complete video sequence from the same scene.*
*Your task is to find the exact temporal position where the query clip appears in the complete video sequence.*
*IMPORTANT CONTEXT:*
*− The query clip shows s consecutive frames from a video sequence*
*− The complete video sequence shows ALL frames from the same scene in chronological order*
*− The query clip appears as a consecutive subsequence somewhere within the complete video sequence*
*− You need to find the exact start and end frame numbers where this subsequence appears*

*IMAGE STRUCTURE:*
*− Images 1−>s: Query video clip (consecutive frames to find)*
*− Images {s+1}−{s+1+video_len}: Complete video sequence (all frames in chronological order)*

*TOTAL IMAGES: {query_count + database_count} images*

*TASK:*
*1. Look at the query clip to understand what sequence you're looking for*
*2. Search through the complete video sequence to find where this exact sequence appears*
*3. The query sequence should appear as consecutive frames in the complete video sequence*
*4. Pay attention to camera movements, object positions, and scene changes to identify the matching sequence*

*FRAME NUMBERING:*
*− The complete video sequence frames are numbered from {min(database_frame_numbers)} to {max(database_frame_numbers)}*
*− You need to return the actual frame numbers from this range*

*RESPONSE FORMAT:*
*Respond with ONLY two numbers separated by a comma: "start_frame,end_frame"*
*− start_frame: The frame number where the query clip begins in the complete video sequence*
*− end_frame: The frame number where the query clip ends in the complete video sequence*

*Example: If the query clip appears at frames 15−19 in the complete sequence, respond: "15,19"*
*Valid frame range: {min(database_frame_numbers)} to {max(database_frame_numbers)}*

---

### A.5.3 SETUPS FOR EGOCENTRIC DEPTH ESTIMATION AT NIGHT.

We evaluate four off-the-shelf monocular depth systems without night-specific fine-tuning. For each, we highlight features pertinent to our setting. (F) denotes support for fisheye egocentric images; (U) denotes undistorted/pinhole images.

1. **Depth Anything V2 (metric). (U)** Foundation MDE model (DPT head with DINOv2 backbone) trained on large-scale synthetic labels plus pseudo-labeled real images. We use the *official metric* checkpoints: *Indoor* (Hypersim-tuned) for indoor frames and *Outdoor* (VKITTI2-tuned) for outdoor frames. Outputs metric depth in meters and is known for strong zero-shot generalization.

2. **StreamVGGT. (U)** A causal/streaming transformer for video geometry that processes frames sequentially with state caching to improve temporal consistency and enable real-time inference. We run it in streaming mode to obtain per-frame depth on egocentric sequences.

3. **Depth Any Camera (DAC). (F)** Zero-shot *metric* depth across diverse camera models via a unified ERP (equirectangular) representation with pitch-aware image-to-ERP conversion and FoV alignment. We use the official release with default settings on our pinhole inputs.

4. **UniK3D. (F)** Universal-camera monocular 3D estimation with a spherical 3D formulation and a learned "pencil-of-rays" camera module, enabling accurate metric depth across pinhole, fisheye, and panoramic views. We run the official model in eval mode; when available, we provide intrinsics for pinhole frames.

| Model | Object Rec. | Text Rec. | Spatial | Scene Seq. | Nav. | Light Rec. | Counting | Non-Common | Overall |
|---|---|---|---|---|---|---|---|---|---|
| *Closed-Source MLLMs* | | | | | | | | | |
| Gemini | 25.94 | 39.39 | 32.43 | 35.47 | 30.77 | 31.97 | 21.88 | 15.15 | 28.34 |
| GPT-4.1 | 25.19 | 54.55 | 35.42 | 28.44 | 27.09 | 35.25 | 20.83 | 16.67 | 27.75 |
| *Open-Source MLLMs* | | | | | | | | | |
| InternVL3-8B | 17.29 | 10.61 | 28.34 | 20.80 | 10.37 | 18.03 | 16.93 | 21.21 | 18.97 |
| Qwen2.5-VL-72B | 15.41 | 16.67 | 28.88 | 21.41 | 7.02 | 12.30 | 10.94 | 21.21 | 17.15 |
| Qwen2.5-VL-7B | 6.77 | 13.64 | 17.98 | 11.93 | 9.03 | 15.57 | 14.06 | 18.69 | 13.26 |
| Qwen2.5-VL-3B | 7.89 | 22.73 | 17.17 | 14.37 | 11.37 | 8.20 | 13.02 | 12.63 | 13.06 |
| GLM-4.1V-9B-Base | 13.16 | 36.36 | 23.71 | 21.71 | 7.02 | 15.57 | 19.27 | 14.14 | 17.69 |
| LLaVA-NeXT-Video-7B | 5.26 | 10.61 | 10.08 | 4.59 | 3.01 | 16.39 | 4.69 | 9.60 | 6.85 |
| VideoLLaMA3-7B | 10.90 | 21.21 | 19.07 | 23.55 | 7.02 | 7.38 | 18.49 | 16.67 | 15.97 |
| *Egocentric MLLMs* | | | | | | | | | |
| EgoGPT | 6.02 | 19.70 | 18.53 | 19.88 | 8.36 | 8.20 | 17.71 | 17.68 | 14.79 |
| *Average across all models* | | | | | | | | | |
| Average | 13.38 | 24.55 | 23.16 | 20.21 | 12.11 | 16.89 | 15.78 | 16.36 | 17.38 |

Table 11: Night-time VQA accuracy (%) per model across all QA categories for EgoNight-Synthetic.

## A.6 More Experimental Results

| Model | Object Rec. | Text Rec. | Spatial | Scene Seq. | Action | Nav. | Light Rec. | Counting | Dyn. Light | Dynamic | Dyn. Count | Avg. |
|---|---|---|---|---|---|---|---|---|---|---|---|---|
| *Closed-Source MLLMs* | | | | | | | | | | | | |
| GPT-4.1 | 24.44 | 33.78 | 41.32 | 30.09 | 38.10 | 27.27 | 38.33 | 24.62 | 30.00 | 13.33 | 25.00 | 31.06 |
| Gemini | 32.22 | 47.30 | 35.54 | 24.78 | 34.92 | 29.29 | 43.33 | 27.69 | 15.00 | 26.67 | 40.00 | 32.67 |
| *Open-Source MLLMs* | | | | | | | | | | | | |
| InternVL3-8B | 16.67 | 17.57 | 33.88 | 24.78 | 22.22 | 21.21 | 20.00 | 22.31 | 25.00 | 6.67 | 15.00 | 22.61 |
| Qwen2.5-VL-72B | 14.44 | 21.62 | 36.36 | 18.58 | 19.05 | 25.25 | 18.33 | 13.85 | 20.00 | 6.67 | 20.00 | 20.99 |
| Qwen2.5-VL-7B | 7.78 | 10.81 | 21.49 | 18.58 | 11.11 | 18.18 | 1.52 | 15.38 | 9.52 | 6.25 | 15.00 | 14.02 |
| Qwen2.5-VL-3B | 11.11 | 8.11 | 23.97 | 13.27 | 11.11 | 16.16 | 10.00 | 15.38 | 5.00 | 13.33 | 10.00 | 14.16 |
| GLM-4.1V-9B-Base | 12.22 | 12.16 | 30.58 | 15.04 | 14.29 | 17.17 | 11.67 | 25.38 | 15.00 | 6.67 | 10.00 | 18.14 |
| VideoLLaMA3-7B | 3.33 | 5.41 | 13.22 | 11.50 | 6.35 | 9.09 | 8.33 | 18.46 | 10.00 | 20.00 | 15.00 | 10.68 |
| LLaVA-NeXT-Video-7B | 8.89 | 5.41 | 18.18 | 12.39 | 12.70 | 15.15 | 11.67 | 13.85 | 0.00 | 13.33 | 20.00 | 12.67 |
| *Egocentric MLLMs* | | | | | | | | | | | | |
| EgoGPT | 9.18 | 3.41 | 19.26 | 16.54 | 6.67 | 7.96 | 10.00 | 14.97 | 0.00 | 0.00 | 10.00 | 11.47 |
| *Average across all models* | | | | | | | | | | | | |
| Average | 13.99 | 16.31 | 27.29 | 18.53 | 17.45 | 18.53 | 17.16 | 19.13 | 12.94 | 11.26 | 18.00 | 18.76 |

Table 12: Night-time VQA accuracy (%) per model across all QA categories for EgoNight-Sofia.

| Models | Object Rec. | Text Rec. | Spatial | Scene Seq. | Action | Nav. | Light Rec. | Counting | Dyn. Light | Dynamic | Dyn. Count | Avg. |
|---|---|---|---|---|---|---|---|---|---|---|---|---|
| *Closed-Source MLLMs* | | | | | | | | | | | | |
| GPT-4.1 | 64.52 | 34.88 | 41.35 | 34.13 | 65.59 | 43.75 | 30.43 | 18.49 | 18.52 | 37.84 | 18.52 | 38.95 |
| Gemini 2.5 Pro | 56.14 | 46.51 | 38.30 | 27.27 | 59.55 | 32.65 | 31.11 | 18.97 | 23.33 | 37.14 | 3.70 | 34.83 |
| *Open-source MLLMs* | | | | | | | | | | | | |
| InternVL3-8B | 40.00 | 27.91 | 18.68 | 14.66 | 36.78 | 20.83 | 6.98 | 9.91 | 6.67 | 37.14 | 7.41 | 20.57 |
| Qwen2.5-VL-72B | 38.18 | 39.53 | 29.67 | 13.79 | 22.99 | 23.96 | 16.28 | 17.12 | 16.67 | 11.43 | 11.11 | 22.07 |
| Qwen2.5-VL-7B | 9.09 | 23.26 | 20.88 | 11.21 | 10.34 | 15.62 | 9.30 | 11.71 | 3.33 | 11.43 | 18.52 | 13.35 |
| Qwen2.5-VL-3B | 14.55 | 13.95 | 9.89 | 13.79 | 19.54 | 20.83 | 6.67 | 10.81 | 3.70 | 17.14 | 6.98 | 13.62 |
| GLM-4V | 22.81 | 30.23 | 21.43 | 15.52 | 28.74 | 9.38 | 19.57 | 17.12 | 6.67 | 37.14 | 14.81 | 19.57 |
| VideoLLaMA3-7B | 20.00 | 11.63 | 15.38 | 9.57 | 10.34 | 7.29 | 9.52 | 9.01 | 0.00 | 17.65 | 7.41 | 10.81 |
| LLaVA-NeXT-Video-7B | 9.09 | 4.65 | 10.99 | 0.00 | 0.00 | 0.00 | 6.98 | 0.00 | 0.00 | 2.86 | 0.00 | 2.86 |
| *Egocentric MLLMs* | | | | | | | | | | | | |
| EgoGPT | 21.13 | 27.08 | 14.14 | 3.42 | 14.61 | 6.25 | 11.11 | 6.25 | 21.62 | 18.92 | 17.86 | 12.44 |
| *Average across all models* | | | | | | | | | | | | |
| Avg. | 29.81 | 25.98 | 22.32 | 14.55 | 27.16 | 18.09 | 14.96 | 12.01 | 10.75 | 22.95 | 10.33 | 19.04 |

Table 13: Night-time VQA accuracy (%) per model across all QA categories for EgoNight-Oxford.

In this section, we show models per QA accuracy on each dataset for EgoNight-Synthetic in Tab. 11, EgoNight-Sofia in Tab. 12, and EgoNight-Oxford in Tab. 13.

## A.7 MORE VISUALIZATION RESULTS

### A.7.1 EGONIGHT-VQA

We further investigate failure reasons by systematically altering illumination, motion blur, and camera noise (sampling quality) in synthetic videos, while keeping scene, annotations, and trajectory fixed in the synthetic dataset. The procedure of generating such difficulty level is described in Appendix. A.1, we calculate the averaged accuracy from Tab. 1 as 18.47% for Easy (moderately dark), 15.88% for Medium (very dark with camera noise), 12.95% for Hard (as dark as medium, with motion blur). From the results, we highlight that: ii) Lower illumination together with camera sensor noise leads to the most significant drop due to loss of contrast and missing fine details. ii) Motion blur further harms performance by causing temporal ambiguity and object shape distortion. iii) These results confirm that night conditions VQA is challenging, highlighting the need for specialized night-egocentric benchmarks. We also visualize more failure cases, and give more analysis in Appendix. A.7.1. Here we provide more failure cases, which compare the day and night VQA output in Fig. 12. This clearly shows the gap between the day and night video understanding. Also, we provide examples of question generated and the corresponding caption to show caption reasoning ability in Fig. 13. Here we provide a more detailed analysis of the failure reason:

- **Extreme Illumination (Counting – Static):** Due to strong red lighting, two doors in the scene become nearly invisible, causing the model to undercount objects.

- **Small Object Disappearance (Text Recognition):** The price tag becomes too small and poorly illuminated at night, making it unreadable and leading to text recognition failure.

- **Spatial Confusion from Limited View (Navigation):** Restricted field of view at night hides key spatial cues (e.g., landmarks, corridor orientation), causing incorrect navigation decisions.

- **Motion Blur (Action Recognition):** Fast hand movement introduces motion blur, making the model misinterpret the content displayed on the screen.

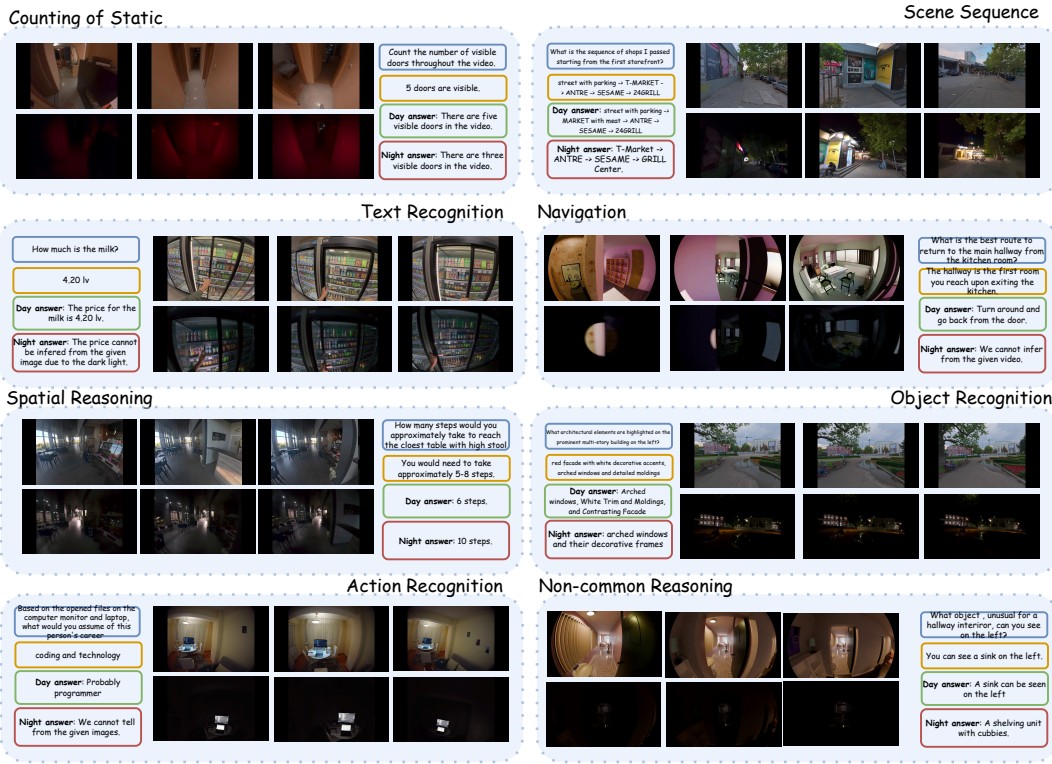

Figure 12: More QA examples with day and night answer produced by the same model.

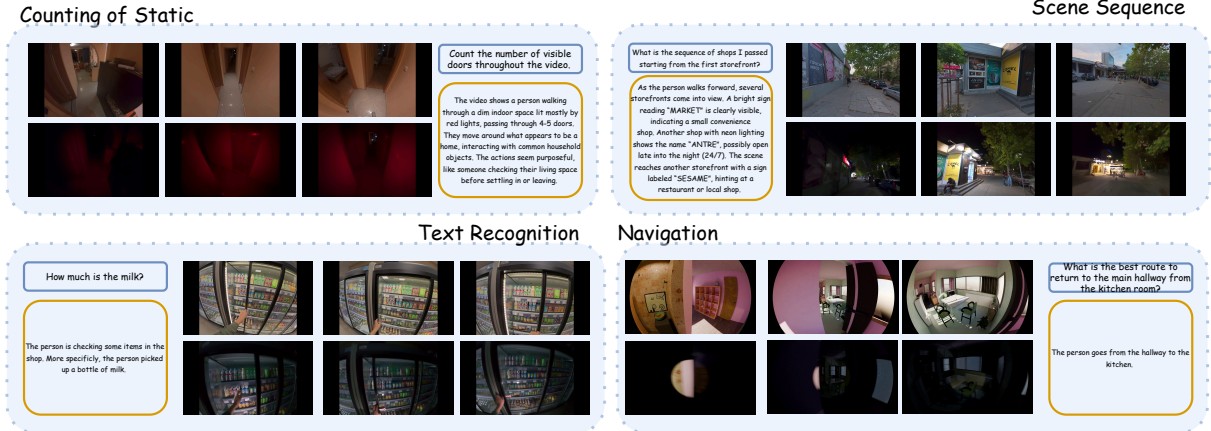

Figure 13: More examples shows the caption together with generated questions.

### A.7.2 DAY-NIGHT CORRESPONDENCE RETRIEVAL

We visualize the qualitative result on one meta sample of Night-to-Day spatial retrieval to better demonstrate the experiment setup and the performance of the benchmarked methods. As shown in Fig. 14, the light condition of the query video clip is drastically different from that of the database clips. Such a difference imposes a great challenge for existing methods in distinguishing the target scenes from the other candidate databases' clips, showing the value of the dataset in the place recognition task.

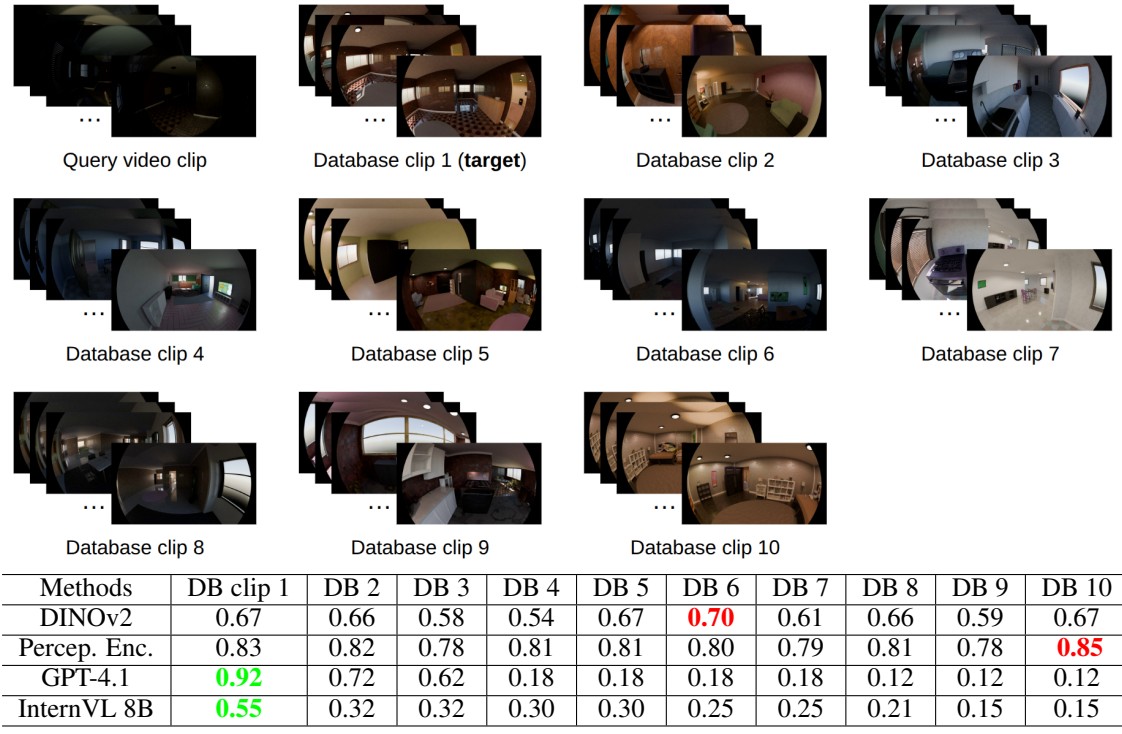

| Methods | DB clip 1 | DB 2 | DB 3 | DB 4 | DB 5 | DB 6 | DB 7 | DB 8 | DB 9 | DB 10 |
|---|---|---|---|---|---|---|---|---|---|---|
| DINOv2 | 0.67 | 0.66 | 0.58 | 0.54 | 0.67 | **0.70** | 0.61 | 0.66 | 0.59 | 0.67 |
| Percep. Enc. | 0.83 | 0.82 | 0.78 | 0.81 | 0.81 | 0.80 | 0.79 | 0.81 | 0.78 | **0.85** |
| GPT-4.1 | **0.92** | 0.72 | 0.62 | 0.18 | 0.18 | 0.18 | 0.18 | 0.12 | 0.12 | 0.12 |
| InternVL 8B | **0.55** | 0.32 | 0.32 | 0.30 | 0.30 | 0.25 | 0.25 | 0.21 | 0.15 | 0.15 |

Figure 14: Qualitative Result on one meta sample of spatial retrieval. The query video clip and the database video clips are visualized in the image. The table below the figure shows the similarity score between the query and the database clips calculated with different methods. The most similar one is in **bold**, and correct retrieval is in **green**, and the incorrect one is in **red**.

### A.7.3 EGOCENTRIC DEPTH ESTIMATION AT NIGHT

We provide additional qualitative results across day–night conditions (Figs. 15, 16, 17, 18). Consistent with the main paper, nighttime is substantially more challenging: low SNR, head-motion blur, extreme dynamic range, color/white-balance shifts, and auto-exposure fluctuations amplify scale ambiguity and erode edge fidelity, leading to over-smoothed surfaces, depth collapse in dark regions, halos around bright point sources, and temporal instability. UniK3D remains the strongest overall in preserving scene structure under these conditions, though performance still degrades under extreme darkness and sparse texture. By contrast, *StreamVGGT* and *DAC* are notably brittle at night, frequently washing out structure, misinterpreting specular highlights, and producing flattened or unstable depth in large low-illumination areas. The effect is most pronounced outdoors in EgoNight-Sofia and EgoNight-Oxford, where wide dynamic range, sparse texture, and point-light saturation further depress accuracy across methods.

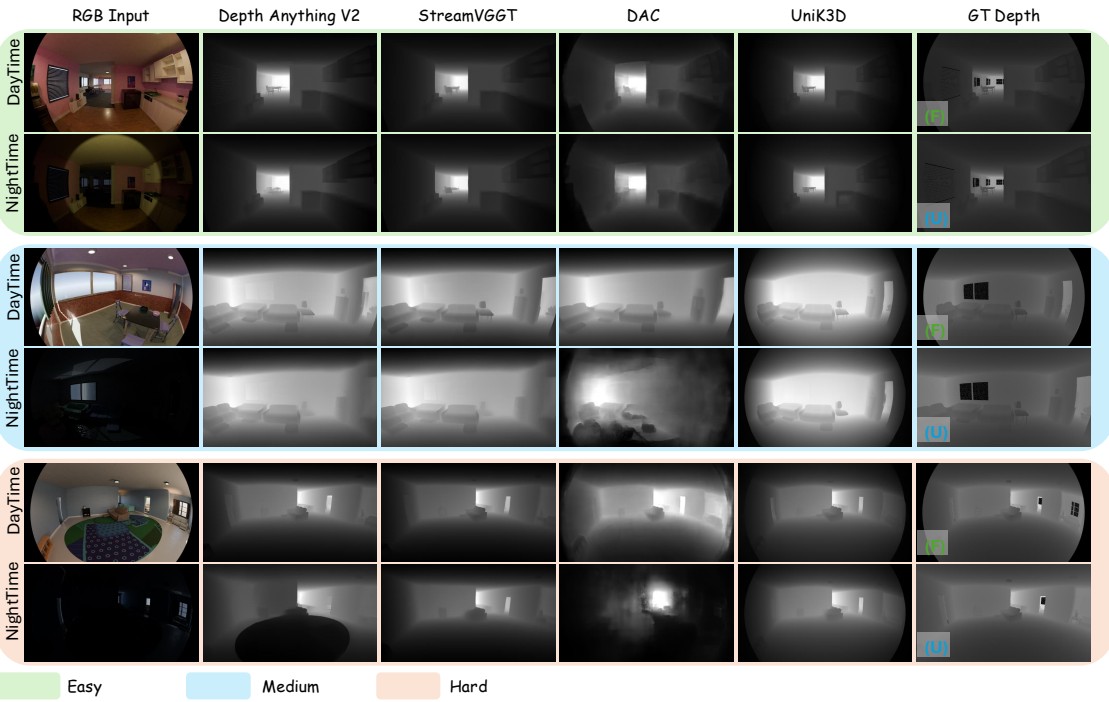

Figure 15: Qualitative results of monodepth estimation in day and night on EgoNight-Synthetic dataset according to different difficulty levels.

## A.8 MORE ANALYSIS

### A.8.1 FAILURECASE ANALYSIS

We further investigate failure reasons by systematically altering illumination, motion blur, and camera noise (sampling quality) in synthetic videos, while keeping scene, annotations, and trajectory fixed in the synthetic dataset. The procedure of generating such difficulty level is described in Appendix. A.1, we calculate the averaged accuracy from Tab. 1 as $18.47\%$ for Easy (moderately dark), $15.88\%$ for Medium (very dark with camera noise), $12.95\%$ for Hard (as dark as medium, with motion blur). From the results, we highlight that: ii) Lower illumination together with camera sensor noise leads to the most significant drop due to loss of contrast and missing fine details. ii) Motion blur further harms performance by causing temporal ambiguity and object shape distortion. iii) These results confirm that night conditions VQA is challenging, highlighting the need for specialized night-egocentric benchmarks. We also visualize more failure cases, and give more analysis in Appendix. A.7.1.

### A.8.2 LIMITATIONS AND FUTURE WORKS

We acknowledge limitations of EgoNight and insights for future research directions in night video understanding. (1) The dataset scale remains modest compared to large-scale vision–language corpora. However, as a testbed, we argue that the current scale of 3,600+ human-verified QA pairs is already sufficient for benchmarking. In future work, we

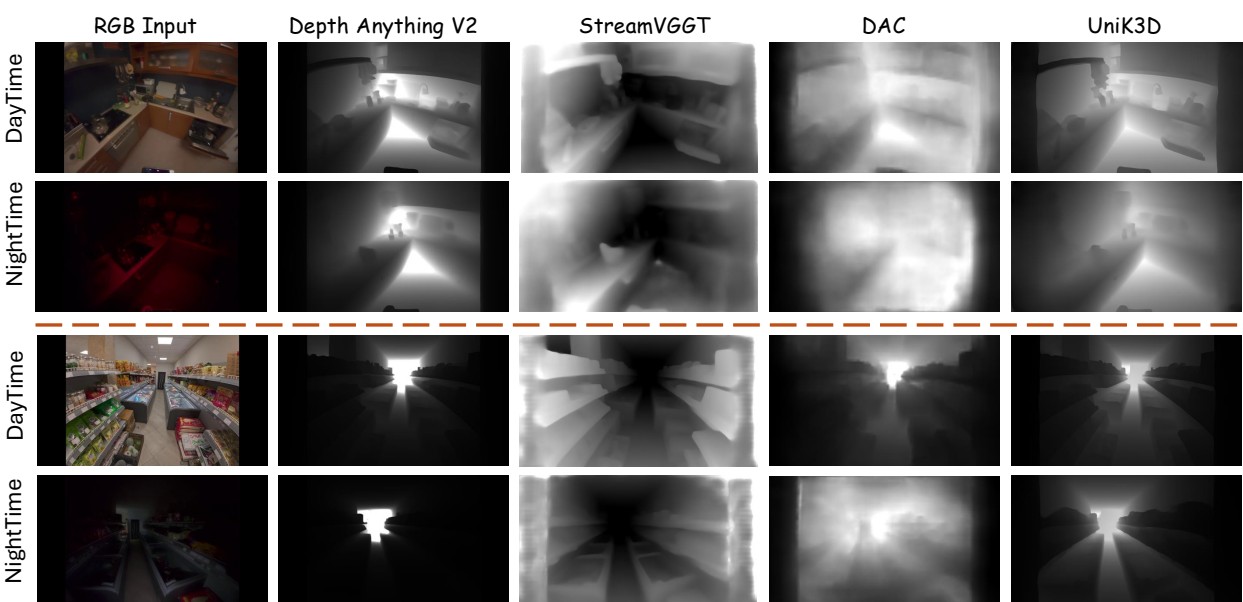

Figure 16: Qualitative results of monodepth estimation in day and night on EgoNight-Sofia dataset indoor part.

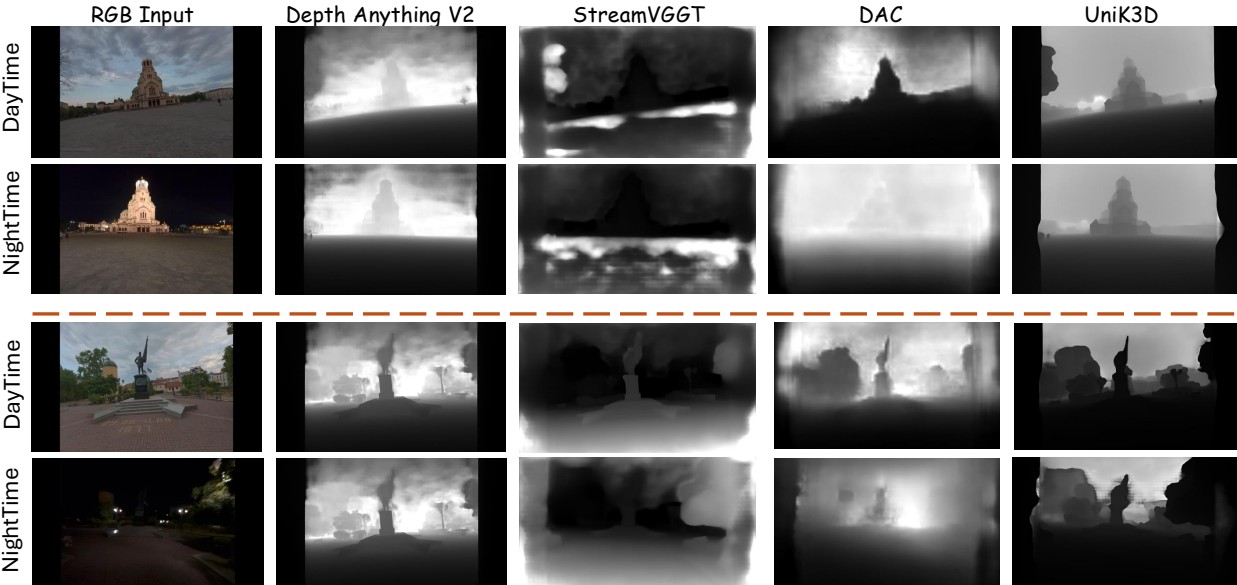

Figure 17: Qualitative results of monodepth estimation in day and night on EgoNight-Sofia dataset outdoor part.

plan to further scale up nighttime videos by synthesizing more data and recording additional real-world footage, which will enable not only benchmarking but also pretraining and fine-tuning to improve MLLM performance.

(2) We show in the main content that fine-tuning on synthetic data improves real world performance. Therefore, we can scale-up synthetic data to build a training set, that can be used to fine-tune the model, thus generalize to real world scenario.

(3) We encourage the community to explore broader research avenues, such as:

- Leveraging unlabeled nighttime or partially annotated video corpora, even if not strictly egocentric;
- Integrating low-level vision techniques for illumination enhancement or robust feature extraction;

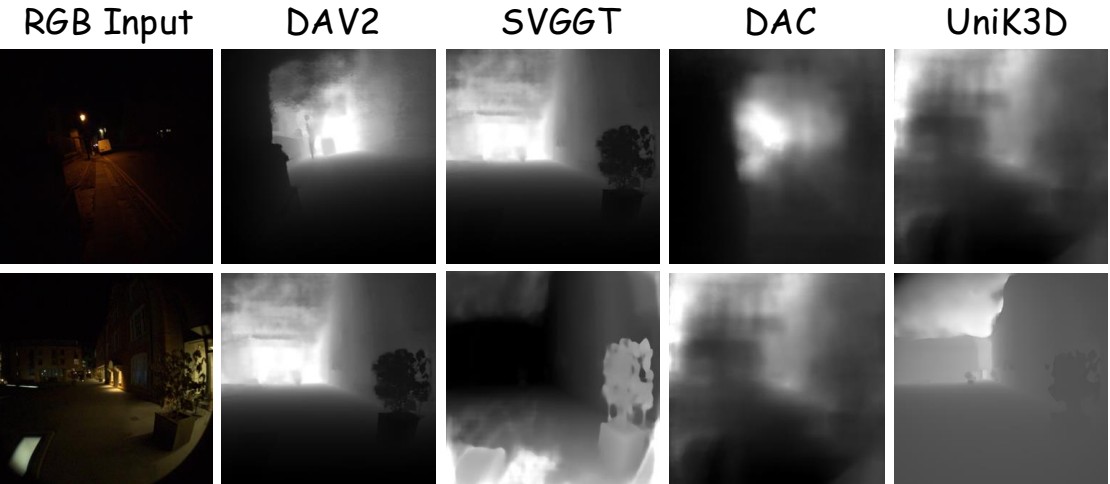

Figure 18: Qualitative results of monodepth estimation in day and night on EgoNight-Oxford dataset, note that DAV2 and SVGGT are shortened for Depth Anything V2 and StreamVGGT respectively.

- Exploring multimodal signal, e.g., depth from EgoNight-Synthetic, to improve low-light understanding;
- Developing training-data-free or lightweight adaptation methods that are more generalizable across MLLMs.

(4) EgoNight primarily focuses on day–night illumination shifts, while other real-world challenges such as weather variations (rain, fog) and extreme camera motion are not covered. We view these as promising directions for future extensions of EgoNight.

### A.8.3 CONTRIBUTION TO THE COMMUNITY

We believe EgoNight will serve as a valuable resource for the research community in several ways. First, it provides the *first benchmark suite* dedicated to egocentric nighttime vision, a long-overlooked but practically critical setting for robust AI assistants. Second, the dataset's unique day–night alignment enables rigorous analysis of illumination effects, offering insights that cannot be obtained from prior egocentric benchmarks. Third, by covering multiple tasks, VQA, day-night correspondence retrieval, and depth estimation, EgoNight provides a comprehensive testbed that can catalyze progress across both perception and reasoning. Finally, with all data, annotations, and evaluation code to be released publicly, EgoNight is designed to be easily accessible, extensible, and reproducible, supporting future research on egocentric vision understanding learning.

### A.8.4 USAGE OF LARGE LANGUAGE MODELS (LLMs)

Our annotation pipeline and benchmark evaluation both leverage large language models (LLMs). For data construction, advanced multimodal LLMs are used to generate initial captions, questions, and pseudo answers, which are then refined by human annotators. This hybrid model–human approach substantially reduces annotation cost while ensuring quality. For evaluation, we adopt the *LLM-as-a-Judge* paradigm to assess the semantic correctness of model outputs against ground-truth answers, following recent practice in egocentric VQA. Beyond annotation and evaluation, we also used LLMs to support paper preparation, such as generating icons for illustration figures and assisting with proof-reading. Importantly, while LLMs serve as practical tools throughout our workflow, all core ideas, dataset design, experiments, and analyses are conceived and conducted independently by the authors.

### A.8.5 ETHIC STATEMENT

All indoor egocentric recordings in EgoNight-Sofia were collected with explicit informed consent, and outdoor data is fully anonymized by blurring faces, license plates, house numbers, and other identifiable details, with audio removed, in compliance with GDPR and privacy standards. Before release, all videos will be verified to contain no personally identifiable information. For EgoNight-Oxford, the subset is derived from the publicly available Oxford Day-and-Night dataset under the BSD-3-Clause license, which permits redistribution and modification with proper attribution. We will retain all required license notices and appropriately acknowledge the original dataset.