# OpenReview forum: "EgoNight: Towards Egocentric Vision Understanding at Night with a Challenging Benchmark"
_ICLR.cc/2026/Conference — ICLR 2026 Poster_

### Official Review · Reviewer_VX41 · 2025-10-14

**Soundness:** 3
**Presentation:** 3
**Contribution:** 3
**Rating:** 6
**Confidence:** 4

**Summary:**

The paper presents EgoNight, a new dataset and benchmark for low-light egocentric video understanding. EgoNight has three subsets, (1) a synthetic video dataset rendered by Blender, where the authors can have more fine-grained control of the scene and the lighting conditions, (2) an aligned day-night real video dataset collected by having video recorders walking around Sofia, Bulgaria at similar speed and route during the day and the night, and (3) a subset of the Oxford day-and-night video dataset. For each subset, a set of questions and corresponding answers are generated for each of the 12 type of questions, and these generated questions and answers are verified by human annotators. In addition to VQA, the benchmark also includes day-night correspondence retrieval, temporal localization, and night-time depth estimation as additional tasks for evaluating the performance of multimodal LLMs under low-light egocentric video conditions.

**Strengths:**

- The paper is well written and easy to understand. The figures are clear and provides a great summarization to the dataset construction pipeline. The paper provides abundant examples from the dataset and both qualitative and quantitative results (some in the supplementary material).
- The new real video recordings in the dataset (especially EgoNight-Sofia) can be a good contribution to the community, especially its aligned day-night recordings captured through the video-guided recording strategy.
- The day–night performance gap illustrated in figure 5 is very significant, highlighting the necessity for a low-light egocentric video dataset and benchmark.
- In addition to VQA, the benchmark also include a few additional tasks that have decent practical value, and allows the user to evaluate the capabilities of MLLMs in low-light egocentric videos more holistically.

**Weaknesses:**

- The paper lacks sufficient discussion regarding the human verification stage. Since most of the ground-truth answers are initially generated from multimodal LLMs, the quality of the human verification stage directly determines the quality of the dataset. Therefore I think the paper should provide more information about the human verification stage, including detailed guidelines to the human annotators and reporting inter-annotator statistics. It would be even better if a subset of the data can be annotated by humans without the help of LLMs to check whether there is bias in the human verification process.
- There are a few ethical concerns regarding the dataset. Details in the ethical concerns section below.

**Questions:**

I wonder whether the authors have did some more controlled experiment with the synthetic data generator to explore why there is such a significant gap in performance between the daytime and nighttime videos. What factors (e.g. illumination, motion blur, location of light source) contribute most to the performance degradation? What are the main failure modes in low light conditions (e.g. spatial confusion, missing small objects)?

**Details Of Ethics Concerns:**

Egocentric recordings in EgoNight-Sofia can expose private information like identifiable faces or license plates. It is unclear from the paper whether sufficient post-processing (e.g. blurring or anonymization) was applied before the data release. The paper also does not discuss IRB approval or GDPR considerations.

---

> ### Author Response · Authors · 2025-11-22
> **Comment about W1, Q1, W2**
>
> # W1: More Discussion on the Human Verification Stage
> Thank you for the helpful comments. We fully agree that the human verification stage is crucial, and we would like to emphasize that we also placed significant effort into ensuring its quality, spending over 200+ annotator hours. Every QA pair was verified by at least one human annotator.
> As detailed in Sec. A.2.1 (Appendix), we provide a clear and structured verification protocol. A simplified version of the annotation tutorial is shown in Fig. 7 (Appendix). It specifies
> * preparation steps
> * how annotators should verify and refine QAs (delete/modify/add)
> * special cases requiring extra care, including the different labeling strategies for paired vs. unpaired QA types
> * post-processing.
>
> We also conducted an onboarding meeting to familiarize annotators with the rules and show several labelling examples, and several mid-stage meetings whenever we identified undesirable labeling patterns through random audits.
>
> **Inter-annotator statistics & the request for a fully human-only subset**: We select randomly VQA pairs from 20 videos out of 90, and add four human annotators without GPT annotation. Here is the average pairwise cosine similarity normalized to (0,1) based on language feature extracted by BLIP-2:
> Average pairwise similarity: 0.8458
> Min pairwise similarity: 0.6711
> Max pairwise similarity: 1.0000
> This shows that human annotators are in general consistent in interpreting scenes. And answers are semantically aligned, even when worded differently.
>
> # Q1: More Analysis Regarding Controlled Experiments.
>
> To better understand the performance degradation in low-light conditions, we conducted controlled experiments varying three key simulation factors: illumination level, sampling quality (noise), and motion blur. As described in Table 4 (Sec. A.1), we define three difficulty levels in our synthetic dataset. We evaluate all models on these three conditions, keeping the semantics, camera path, and question annotations identical.
>
> | Difficulty | Light Condition                                     | Render Sampling                  | Motion Blur | Accuracy (%) |
> | ---------- | --------------------------------------------------- | -------------------------------- | ----------- | ------------ |
> | Easy       | Moderately dark (105° spotlight or multiple lights) | High (4096 spp) – low noise      | No          | 18.47        |
> | Medium     | Very dark (40–50° spotlight, few lights)            | Low (512 spp) – noticeable noise | No          | 15.88        |
> | Hard       | Same as medium                                      | Low (512 spp) – noticeable noise | Yes         | 12.95        |
>
>
> We observe that the illumination decrease causes the largest drop from 18.47 to 15.88. This is due to reduced scene visibility, loss of object/shadow contrast, and disappearance of fine details. Also, motion blur further harms performance from 15.88 to 12.95, which because of temporal ambiguity and shape distortion.
>
> We visualized more failure cases in Appendix Figure 12, we provide one representative failure case for each task category and summarize the main causes. Here we provide a more detailed analysis of the failure reason:
> * **Extreme Illumination (Counting – Static)**: Due to strong red lighting, two doors in the scene become nearly invisible, causing the model to undercount objects.
>
> * **Small Object Disappearance (Text Recognition)**: The price tag becomes too small and poorly illuminated at night, making it unreadable and leading to text recognition failure.
>
> * **Spatial Confusion from Limited View (Navigation)**: Restricted field of view at night hides key spatial cues (e.g., landmarks, corridor orientation), causing incorrect navigation decisions.
>
> * **Motion Blur (Action Recognition)**: Fast hand movement introduces motion blur, making the model misinterpret the content displayed on the screen.
>
> # W2: Ethics Concerns.
> Thank you for raising this concern. For all indoor recordings, participants provided explicit informed consent for data collection and research use. For outdoor videos, we perform comprehensive anonymization, including blurring faces, license plates, house numbers, and any other identifiable details. Besides, audio will also be removed. Before releasing EgoNight-Sofia, we will ensure that all videos are fully anonymized and contain no personally identifiable information (PII), in accordance with standard privacy and GDPR.

---

> > ### Comment · Reviewer_VX41 · 2025-11-23
> >
> > I thank the authors for their thoughtful response. Although I would still like the paper to go through an independent ethics review, the authors' responses (and the additional results provided in responses to other reviewers) have resolved my technical concerns and I have increased my score accordingly.

---

> > > ### Author Response · Authors · 2025-11-24
> > >
> > > Thank you very much for taking the time to re-evaluate our work. We truly appreciate your careful consideration, and we are grateful that our responses and additional results were helpful in addressing your technical concerns. We will keep your suggestion regarding an independent ethics review in mind and will ensure that all privacy-related issues are handled responsibly before releasing the dataset. Thank you again for your detailed feedback and for improving your score.

---

### Official Review · Reviewer_3vts · 2025-10-29

**Soundness:** 2
**Presentation:** 2
**Contribution:** 2
**Rating:** 4
**Confidence:** 4

**Summary:**

This paper introduces EgoNight, the first comprehensive benchmark dataset designed to evaluate egocentric vision understanding, particularly for Multimodal Large Language Models (MLLMs), under challenging nighttime and low-light conditions. The authors identify that existing egocentric datasets focus almost exclusively on well-lit daytime scenarios, ignoring a critical operational domain for real-world applications like personal assistants. The core contribution is the creation of a benchmark suite featuring day-night aligned videos, which is crucial for two reasons: (i) It enables a rigorous and fair comparison of model performance, isolating the performance drop caused by illumination changes. (ii) It facilitates a novel "day-augmented auto-labeling" pipeline, where the clear daytime videos are used to help generate high-quality annotations for the ambiguous nighttime videos.

The EgoNight dataset is sourced from three streams:
- EgoNight-Synthetic: 50 perfectly aligned pairs from a Blender simulation.
- EgoNight-Sofia: 20 real-world pairs captured with a novel video-guided recording protocol to ensure "decent alignment".
- EgoNight-Oxford: 20 unaligned nighttime videos from an existing dataset to test generalization.

Upon this data, the authors build three benchmarks:
- EgoNight-VQA (the core task): 3,658 human-verified question-answer pairs spanning 12 diverse QA types, including novel tasks like "lighting recognition" and "non-common-sense reasoning".
- Day-Night Correspondence Retrieval: Evaluates spatial and temporal matching across illumination conditions.
- Egocentric Depth Estimation: Uses the synthetic data's ground-truth depth to test models at night.

Extensive experiments on over 10 state-of-the-art MLLMs (including GPT-4.1 and Gemini 2.5 Pro) reveal a substantial performance drop from day to night (e.g., 32.8% on Synthetic, 25.0% on Sofia), demonstrating that current models are not robust to low-light conditions and validating the difficulty and necessity of this new benchmark.

**Strengths:**

- The paper tackles a critical and previously overlooked gap in egocentric vision. The benchmark (EgoNight) is the first of its kind and will be a valuable community resource.

- The paper's use of both perfectly-aligned synthetic data and realistically-aligned real-world data provides a robust and comprehensive foundation for evaluation.

- High-Quality Annotation Pipeline: The "day-augmented auto QA generation" followed by over 300 hours of human verification ensures the reliability of the core VQA benchmark, which is especially difficult for low-visibility data.

- Comprehensive Evaluation: The paper presents not just one VQA task, but a suite of three benchmarks (VQA, Retrieval, Depth).

**Weaknesses:**

- The scale of real-world data is pretty small. Only 20 day–night pairs in EgoNight-Sofia dataset.

- The quality of simulation data is uncertain. Both qualitative and quantitative studies are missing.

**Questions:**

- Could the authors provide more analysis and examples of the synthetic data?

---

> ### Author Response · Authors · 2025-11-22
> **Comment about W1, W2, Q1**
>
> Thank you for the feedback and positive assessment of the benchmark’s significance towards real-world applications, task design, annotation pipeline, and evaluation. We address each concern below.
>
> # W1: Scale of Real-World Data.
> We acknowledge that EgoNight-Sofia contains only 20 day–night video pairs; however, as a testbed, this scale is appropriate and comparable to other egocentric benchmarks.
> * In practice, the usefulness of a benchmark is determined by the number and diversity of VQA pairs, not merely the count of raw videos. For example, as summarized below, EgoLife includes only 6 videos and EgoTempo 40, while EgoNight-Sofia/EgoNight provide 20/90 videos and 813/3658 VQA pairs, within a practical range.
> | Benchmark              | Conference | # Videos | # VQA Pairs |
> | ---------------------- | ---------- | -------- | ----------- |
> | EgoLife                | CVPR 2025  | 6        | 6,000       |
> | EgoTempo               | CVPR 2025  | 40       | 500         |
> | EgoNight-Sofia (Ours)  | –          | 20       | 813         |
> | EgoNight-Oxford (Ours) | –          | 20       | 816         |
> | EgoNight (Ours)        | –          | 90       | 3,658       |
>
> * As shown in Fig. 1, Sec. 3.1, and Sec. A.1, each EgoNight-Sofia pair was carefully recorded and intentionally varied across scenes (apartments, offices, streets, neighborhoods, tourist sites, grocery stores, outdoor fitness areas). This ensures broad coverage and natural diversity across the resulting VQA annotations.
> * Most importantly, Tab. 1 shows that existing models exhibit a large performance spread on EgoNight-Sofia (≈10%–40%), demonstrating that the dataset is sufficiently challenging and effective for distinguishing model capabilities.
> * In addition to EgoNight-Sofia, EgoNight-Oxford should also be regarded as real-world data. Together, the ratio of synthetic to real-world samples is roughly balanced (videos: 50:40; VQA pairs: 2029:1629). Moreover, Tab. 1 shows that the relative ranking of models remains consistent across all subsets and difficulty tiers, indicating that EgoNight-Synthetic, EgoNight-Sofia, and EgoNight-Oxford each provide reliable and coherent evaluations of model performance.
>
> # W2&Q1: More Examples and Analysis of the Synthetic Data.
> Thanks. We clarify that:
> * The details of EgoNight-Synthetic Construction is given in Sec A.1, Appendix, L706-L728, which states each step used for constructing the synthetic dataset. We highlight that: a) all the important steps e.g., edit the light source, record camera trajectory to mimic human are done by human annotators, which ensures the quality of the synthesized data; b) we carefully design three different task levels via applying various sample size, light conditions, and motion blur, to mimic the real low-lighting scenarios as much as possible.
> * **Qualitative examples** of the synthetic data can be found in Fig.6, Appendix, and Fig.9, Appendix. From Fig.6, we show 3 paired day-night frames with its corresponding multimodal data.  From Fig.9, for each QA type, we demonstrate one clip (in frames) example with its QA examples.  Complete videos and QA annotations will be released upon acceptance.
> * **Quantitative analysis** of synthetic data can be drawn from:
>     * the comparison results among easy, medium, and hard  (Tab.1) validate the different levels proposed;
>     * We calculate the pearson correlation of the average score per-model shown in Appendix Tabs. 9–11 between synthetic and sofia (0.9359 with p-value 6.847×10⁻⁵), synthetic and Oxford (0.8588 with p-value 0.001462). These strong and statistically significant correlations indicate that performance on synthetic data is highly predictive of performance on real-world data, further validating its representativeness.
> * We conducted more analysis on synthetic data, more specifically, to further quantitatively assess the quality of our synthetic dataset, we performed SFT to finetune the Qwen2.5-VL-7B model using only the synthetic data and evaluate it on the real dataset. As shown in the table below, the performance improves consistently, with an average gain of about 6%. This demonstrates that our synthetic data can effectively enhance model performance in real-world scenarios.
> | Model                       | Object | Text  | Spatial | SceneSeq | Navigation | Light | Counting | Average |
> | --------------------------- | ------ | ----- | ------- | -------- | ---------- | ----- | -------- | ------- |
> | Qwen2.5-VL-7B               | 7.78   | 10.81 | 21.49   | 18.58    | 18.18      | 1.52  | 15.38    | 14.83   |
> | Qwen2.5-VL-7B-Synthetic-SFT | 14.44  | 24.32 | 23.43   | 20.18    | 26.26      | 20.00 | 16.15    | 20.57   |

---

> > ### Author Response · Authors · 2025-11-26
> >
> > Dear Reviewer,
> >
> > Understanding your busy schedule, we wanted to check if the revisions and details provided have addressed your concerns. Please let us know if any issues remain or further clarification is needed. Your feedback is invaluable to us for improving our work.
> >
> > Thank you for your attention.

---

> > > ### Comment · Reviewer_3vts · 2025-11-27
> > >
> > > I appreciate the authors’ responses. Most of my concerns have been addressed to some extent, and I am inclined to accept this paper.

---

> > > > ### Author Response · Authors · 2025-11-28
> > > >
> > > > Thank you for the recognition and dedication. And we are happy to see that our response address your concerns. Thanks again for your valuable comments to help us improve our paper. We are happy to address your concerns if further clarification is needed!

---

### Official Review · Reviewer_Cyx7 · 2025-11-03

**Soundness:** 3
**Presentation:** 3
**Contribution:** 2
**Rating:** 4
**Confidence:** 4

**Summary:**

This paper introduces EgoNight, a new benchmark suite designed to evaluate egocentric vision understanding in challenging nighttime or low-light conditions. The authors identify this as a critical but overlooked area, as most existing benchmarks focus on daytime scenarios.
The primary contributions are threefold:
1. A Novel Dataset: The EgoNight dataset combines three sources: a synthetic dataset with perfectly aligned day-night video pairs (EgoNight-Synthetic), a real-world dataset with carefully recorded, temporally aligned day-night pairs (EgoNight-Sofia), and an unaligned real-world night video dataset for diversity (EgoNight-Oxford). The day-night alignment is a core feature, enabling controlled analysis of model performance under different illumination.
2. A Comprehensive Benchmark Suite: The main task is Visual Question Answering (VQA), featuring 3,658 human-verified QA pairs across 12 diverse types, including novel tasks like lighting recognition and non-common-sense reasoning. The suite is complemented by two auxiliary tasks: day-night correspondence retrieval and egocentric depth estimation.
3. Some Empirical Analysis: The authors evaluate a wide range of state-of-the-art Multimodal Large Language Models (MLLMs), including closed-source models like GPT-4.1 and Gemini. The results consistently show a substantial performance drop from day to night across all models and tasks, highlighting that current systems are not robust to illumination changes.

**Strengths:**

1. Addresses a Relevant Problem: The paper highlights the practical and important challenge of model robustness to illumination changes, a common failure mode for vision systems in real-world applications.
2. Significant Data Collection Effort: The authors have clearly invested a substantial amount of effort in collecting and annotating a new dataset, particularly in creating the temporally aligned real-world videos (EgoNight-Sofia). This resource, if made public, could be of some use to the community.

**Weaknesses:**

1. Limited Novelty: The work feels incremental. It takes the well-established paradigm of egocentric VQA and applies it to the nighttime domain. This is more of a domain generalization study on a new dataset than a novel research contribution. Prior work like the Oxford Day-and-Night dataset already provided a basis for nighttime egocentric vision, and benchmarks like EgoCross have already explored cross-domain challenges for egocentric VQA. The current work is a straightforward combination of these existing concepts.
2. Potential for Dataset Contamination and Bias: The VQA pairs were bootstrapped using GPT-4.1. This creates a significant risk that the dataset is "contaminated" with the stylistic and factual biases of the model used to generate it. Consequently, the benchmark may unfairly favor models that are architecturally or stylistically similar to GPT-4.1, making it an unreliable tool for impartially assessing a diverse set of MLLMs.
3. Flawed and Unreliable Evaluation: The use of an LLM-as-a-Judge for open-ended VQA is a major methodological weakness. This evaluation strategy is not yet considered robust or unbiased in the research community. Without calibration against human judgments, the reported accuracy numbers are not trustworthy and cannot be reliably used to compare models. This undermines the paper's core empirical claims.
4. Presentation Can Be Further Addressed: The paper suffers from some grammatical errors and lacks the polish expected for a top-tier publication. The writing style is often convoluted, making it difficult to parse key details of the data collection and annotation pipeline.

Here are some mistakes i listed, but more:

Page 2, Line 70-71: "This motivates us to investigate egocentric vision at night, with an emphasis on complex scene understanding and reasoning tasks."
Issue: The sentence structure is a bit convoluted. "With an emphasis on" feels like an add-on. A clearer phrasing might be: "This motivates us to investigate egocentric vision at night, focusing on complex scene understanding and reasoning tasks."

Page 2, Line 74-75: "A central challenge in constructing such a benchmark lies in obtaining suitable video sources that capture the characteristics of nighttime environments, as well as developing annotation methods that ensure high labeling quality."
Issue: The phrase "as well as developing" creates a slightly unbalanced structure. A more parallel construction would be better: "...lies in obtaining suitable video sources... and developing annotation methods..."

Page 2, Line 83-84: "However, in practice, collecting perfectly aligned day-night pairs in the real world is highly nontrival."
Issue: "Nontrival" is not a standard English word. It should be "non-trivial." This is a clear spelling/typographical error.

Page 5: Starting from 3.2, there's always a space at the beginning of the paragraph, which is not aligned with the ICLR template.

**Questions:**

Please address the weakness in the above. Some additional questions:

1. The paper's main argument rests on its novelty as the "first comprehensive benchmark for nighttime egocentric vision." However, given existing work like the Oxford Day-and-Night dataset, could you clarify what makes your contribution fundamentally different, rather than just an extension in scale and task diversity?

2. Regarding the scope of the benchmark: The paper presents EgoNight as a suite for evaluating existing models. However, the true value of a new benchmark often lies in its ability to drive the development of new models. Have the authors considered creating and releasing a dedicated training set? Providing a training split would fundamentally increase the benchmark's impact, allowing the community to build and fine-tune models specifically for the challenges of nighttime egocentric vision. Without a training set and a baseline evaluation of a model fine-tuned on it, the novelty and contribution of the work could be seen as limited to just a test set.

3. The paper's analysis is too simple. While it successfully demonstrates that models fail in low-light conditions, it provides almost no insight into why they fail or how the community should go about fixing them. The key takeaway is a problem statement, but it leaves researchers with no clear direction. For example, is the core issue a lack of diverse, low-light data in the models' massive pre-training datasets, or can this deficit be effectively overcome with targeted fine-tuning (post-training) on a smaller, specialized dataset like EgoNight? The paper does not investigate this. The analysis in Figure 5b shows that perception-oriented tasks (e.g., object/text recognition) degrade more severely than reasoning tasks (e.g., navigation). This is a good observation, but the paper stops there. It fails to diagnose whether this is due to the vision encoder failing to extract meaningful features from noisy pixels (an issue with the pre-trained foundation) or the language model's inability to handle uncertain visual inputs (an issue that might be solved via fine-tuning). The analysis simply presents the final scores without diagnosing the root cause of the failure.

**Details Of Ethics Concerns:**

While EgoNight-Synthetic is self-created and EgoNight-Sofia is self-collected. Do the authors have the legal and ethical right to redistribute a modified portion of the Oxford dataset? This is not stated in the paper. Could the authors please carify?

---

> ### Author Response · Authors · 2025-11-22
> **Part 1: Comment about W1&Q1**
>
> We thank the reviewer for recognizing the importance of our topic, dataset, and the valuable feedback to help us improve our paper. We have revised our paper based on your feedback. We detail our response below and please kindly let us know if our response addresses your concerns.
>
> # W1 & Q1: Limited Novelty and Comparison against Oxford Day-and-Night, EgoCross
> Thanks, while EgoNight is related to domain generalization (as noted in Sec. 2.3), we believe our contribution goes substantially beyond a simple extension of existing datasets such as Oxford Day-and-Night or prior benchmarks like EgoCross. We highlight the key differences below.
>
> ## 1. Fundamental differences from Oxford Day-and-Night
> Although Oxford Day-and-Night contains nighttime egocentric videos, its design and motivation differ fundamentally from ours:
> * **Lack of aligned day–night pairs.** As discussed in L130–131 and Sec. 2.1, Oxford Day-and-Night does not provide spatially–temporally aligned day/night counterparts. In contrast, EgoNight introduces aligned day–night video pairs, which are critical for:
>     * cleanly isolating illumination effects.
>     * enabling high-quality annotation via day-augmented QA generation
>     * quantifying day-to-night performance degradation.
> This alignment is a major contribution and is not available in any prior work.
> * **Different task focus and required perception depth.** Oxford Day-and-Night was primarily designed for 3D tasks (e.g., novel view synthesis), not for VQA-based understanding. EgoNight, by contrast, focuses on high-level perception and reasoning, which requires richer semantic grounding. We also spent considerable time and effort to annotate the Oxford Day-and-Night video to generate VQA pairs—this is not provided in the original dataset. Thus, EgoNight differs from Oxford across motivation, metadata structure, and task formulation.
>
> ## 2. EgoNight is not a subset of EgoCross
>
> As noted in L158–161, both EgoCross and EgoNight touch on domain generalization but target distinct problem spaces:
> * EgoCross explores long-tail, specialized domains (surgery, industry, extreme sports, animal perspectives).
> * EgoNight focuses on nighttime, a common and ubiquitous scenario in daily life, yet one of the most challenging due to severe illumination degradation.
>
> In the vision community, low-light vision[r1] itself is considered an independent research direction, not simply a sub-case of domain generalization. Our benchmark fills this gap specifically for egocentric understanding, which has not been addressed before.
>
> ## 3. EgoNight contributes multiple novel components beyond the dataset itself
>
> Beyond being the first comprehensive benchmark for nighttime egocentric VQA, EgoNight introduces several technical contributions:
>
> * A paired day–night video acquisition strategy (L210–218), which enables spatial-temporal alignment and robust evaluation across illumination conditions.
> * A three-stage day-augmented QA generation pipeline (Fig. 2, Sec. 3.2) tailored for nighttime videos.
> * A large-scale, high-quality annotation effort (3600+ QA pairs with 300+ hours of human verification; L296–302).
> * A multi-task nighttime evaluation suite(Sec. 3.3), including:
>     * VQA
>     * day–night correspondence retrieval,
>     * egocentric depth estimation at night
> * Extensive experiments (Sec. 4) revealing performance degradation patterns unique to nighttime egocentric vision.
>
>
> Taken together, these contributions form a new and practically meaningful benchmark rather than a simple extension of existing datasets.
>
> [r1] Zhao, Qian, et al. "Deep Learning for Low-Light Vision: A Comprehensive Survey." IEEE Transactions on Neural Networks and Learning Systems (2025).

---

> > ### Author Response · Authors · 2025-11-22
> > **Part 2: Comment about W2, W3, Q2**
> >
> > # W2: Potential for Dataset Contamination and Bias from GPT-4.1
> >
> > Thank you for raising this concern. While GPT-4.1 is used to bootstrap the initial QA generation, every QA pair in the final dataset has undergone human verification, following the protocol described in Sec. A.2.1. During this stage, annotators carefully checked, corrected, removed, or rewrote QA pairs as needed. On average, the verification for a single video required approximately 1.5–2 hours of human work, indicating substantial human involvement and ensuring high-quality annotations.
> >
> > To further examine the concern regarding potential stylistic bias toward GPT-generated answers, we divided all QA annotations into two groups:
> >
> > (a) Answers verified but not modified by humans (GPT-like style)
> >
> > (b) Answers modified or fully created by humans (human style)
> >
> > | Group                   | Portion | GPT-4.1 Evaluation Accuracy |
> > | ----------------------- | ------- | --------------------------- |
> > | (a) GPT-style answers   | ~40%    | 26.73%                      |
> > | (b) Human-style answers | ~60%    | 27.87%                      |
> >
> >
> > The scores are comparable, showing no clear preference for GPT-style wording. This suggests that LLM-as-a-Judge evaluates semantic correctness rather than stylistic similarity, reducing concerns of bias toward GPT-like expressions.
> >
> > # W3: Flawed and Unreliable Evaluation from LLM-as-a-Judge.
> > We agree that LLM-based evaluation is not perfectly reliable. However, we would like to clarify the following:
> >
> > * **LLM-as-a-Judge is widely adopted and accepted in egocentric VQA.** In the area of egocentric VQA, using LLM-as-a-Judge has become a common and accepted practice for open-ended answer evaluation. Representative benchmarks that actively employ it include EgoCross, EgoTempo, and EgoLife, among others.
> >
> > * **Our benchmark exhibits stable and meaningful evaluation behavior.**
> > Despite using LLM-as-a-Judge, our results in Table 1 exhibit clear and consistent trends:
> >     *  model performance decreases as the difficulty level increases, and
> >     * models maintain similar relative rankings across datasets and difficulty tiers.
> >  These consistent trends indicate that the benchmark produces stable and meaningful evaluations rather than arbitrary scores.
> >
> >
> > * **Human validation of LLM-as-a-Judge.** Following the reviewer’s concern, we conducted an additional human validation study. We randomly sampled 300 triplets of (question, model-generated answer, ground truth) from all evaluated VLMs and asked human annotators to verify whether the scores assigned by LLM-as-a-Judge were correct. The results show that the LLM-as-a-Judge aligns with human judgment 95.67% of the time.  This high agreement rate demonstrates that the LLM-based evaluation is highly aligned with human judgment and supports its reliability for our benchmark. We added this part to the appendix page 21 Sec. A.5.1 line 1130-1137. And add reference from the main paper at page7 Sec. 4.1 line 367, denoted using blue color.
> >
> > # Q2: The Scope of the Benchmark.
> > Thank you for the suggestion. We fully agree that the long-term value of a benchmark lies in its ability to drive new model development, and this is precisely what motivated EgoNight. We indeed considered releasing an additional training set to make model development easier. However, we ultimately decided to keep EgoNight strictly as a testbed for two main reasons:
> >
> > * While adding more labeled training data could improve accuracy, it is not the only promising direction for advancing nighttime egocentric understanding. We hope EgoNight will inspire broader research avenues, such as:
> >
> >     * Leveraging unlabeled nighttime or partially annotated video corpora, even if not strictly egocentric;
> >
> >     * Integrating low-level vision techniques for illumination enhancement or robust feature extraction;
> >
> >     * Exploring multimodal signals, particularly depth from EgoNight-Synthetic, to improve low-light understanding;
> >
> >     * Developing training-data-free or lightweight adaptation approaches, which are more generalizable across MLLMs.
> >
> >     These directions are equally meaningful and may lead to more innovative and transferable solutions, rather than simply scaling supervised data.
> >
> > * Providing a large, EgoNight-style training set at this early stage may yield easy performance gains but also risks overfitting to our data formulation, thereby limiting innovation. Instead, leaving the training aspect open to the community encourages researchers to explore fundamentally new ideas rather than converging prematurely on a single predefined solution. We believe this will lead to a richer variety of methods and faster progress in nighttime egocentric understanding.

---

> > > ### Author Response · Authors · 2025-11-22
> > > **Part 3: Comment about Q3 (first part)**
> > >
> > > # Q3: More in-depth analysis.
> > > Thanks and we value this suggestion a lot, to provide more in-depth analysis, we further conduct several invesitagations.
> > >
> > > ## 1. Could finetuning help to enhance performace?
> > > To perform this, we first divide EgoNight into 70% training and 30% test splits. We fine-tune Qwen2.5-VL-7B using supervised fine-tuning (SFT) under three configurations:
> > > * Full Fine-Tuning: both vision encoder and LLM are trainable.
> > > * Finetune Encoder Only: only the vision encoder is fine-tuned.
> > > * Finetune LLM Only: only the LLM is fine-tuned.
> > > The accuracy results are summarized a below:
> > > | Dataset               | Qwen7B (Base) | Finetune Encoder Only | Finetune LLM Only | Full Finetune |
> > > | --------------------- | ------------- | --------------------- | ----------------- | ------------- |
> > > | Synthetic             | 23.23         | 29.74                 | 35.50             | **36.25**     |
> > > | Real (Sofia + Oxford) | 16.40         | 20.92                 | 22.26             | **25.61**     |
> > >
> > > From the results, we found that:
> > > * Fine-tuning on EgoNight substantially improves model performance, indicating that introducing EgoNight-style training data effectively adapts models to nighttime scenarios.
> > > * Both vision-encoder tuning and LLM tuning independently contribute to performance gains. Interestingly, fine tuning the LLM only yields a larger improvement, suggesting that visual representation is not the only bottleneck. LLM fine-tuning is also important since it aligns uncertain vision features to language space.
> > > * Full fine-tuning consistently outperforms partial fine-tuning, demonstrating that EgoNight demands both strong visual extraction ability and well visual-language alignment.
> > > ## 2. How Perception vs. Reasoning-Oriented tasks benefit from finetuning?
> > > To further dive into the impact of finetuning, we compare perception-oriented tasks (Object, Text Recognition) and reasoning-oriented tasks (Navigation, Counting) accuracy:
> > > | Task       | Qwen7B (Base) | Finetune Encoder Only | Finetune LLM Only |
> > > | ---------- | ------------- | --------------------- | ----------------- |
> > > | Object     | 8.435         | 34.718                | **35.855**        |
> > > | Text       | 18.440        | 49.890                | **50.988**        |
> > > | Navigation | 17.870        | 19.495                | **19.918**        |
> > > | Counting   | 16.558        | 16.945                | **24.275**        |
> > > From the results, we observe that:
> > > * Perception-oriented tasks (Object, Text Recognition)  are easier to be enhance (improved by 2-4 times) by finetuning than reasoning-oriented tasks (Navigation, Counting), which indicates the visual representation is easier to be learned by adapting to training data than reasoning.
> > > * Finetuning vision encoders improves perception-oriented tasks by 2-4 times, however it doesn't improve reasoning-oriented tasks clearly. This reveals the huge challenges posed by the reasoning-oriented tasks.
> > > * Finetuning the language model also improves both perception-oriented tasks and reasoning-oriented tasks, with larger performance boost observed for those perception-oriented tasks. This indicates that the performance boost is mainly from aligning the uncertain vision feature space to the language space, whereas the enhancement of true reasoning ability remains quite limited.
> > >
> > > ## 3. Could synthetic data help real data?
> > > Since synthetic data is easier to be collected than real data, we are wondering could synthetic data also work as training data to help models adapt to real data? To answer this, we fine-tune Qwen2.5-VL-7B using only synthetic data, then evaluate on real-world (Sofia + Oxford) videos.
> > > | Model                       | Accuracy |
> > > | --------------------------- | ---------------- |
> > > | Qwen2.5-VL-7B               | 14.83            |
> > > | Qwen2.5-VL-7B-Synthetic-SFT | **20.57**        |
> > > We found that training purely on synthetic data leads to a ~6% absolute improvement on real-world videos, demonstrating that simulator-generated data effectively transfers to real egocentric night scenarios.
> > >
> > > ## 4. Controlled Factor Analysis (Failure Diagnosis)
> > > We further investigate failure reasons by systematically altering illumination, motion blur, and camera noise (sampling quality) in synthetic videos, while keeping scene, annotations, and trajectory fixed. Here is the accuracy:
> > > | Difficulty Condition | Accuracy |
> > > | -------------------- | ------------- |
> > > | Normal               | 18.47         |
> > > | ↓ Illumination       | 15.88         |
> > > | + Motion Blur        | 12.95         |
> > > From the results, we highlight that: 1) Lower illumination together with camera sensor noise leads to the most significant drop due to loss of contrast and missing fine details. 2) Motion blur further harms performance by causing temporal ambiguity and object shape distortion. 3) These results confirm that night conditions VQA is challenging, highlighting the need for specialized night-egocentric benchmarks like EgoNight.

---

> > > > ### Author Response · Authors · 2025-11-22
> > > > **Part 4: Comment about Q3 (second half), W4, Ethics Concerns**
> > > >
> > > > # Q3 (continue)
> > > > ## 5. More Failure Case Analysis
> > > > We visualized more failure cases in Appendix Figure 12, we provide one representative failure case for each task category and summarize the main causes. Here we provide a more detailed analysis of the failure reason:
> > > >
> > > > * **Extreme Illumination (Counting – Static)**: Due to strong red lighting, two doors in the scene become nearly invisible, causing the model to undercount objects.
> > > >
> > > >
> > > > * **Small Object Disappearance (Text Recognition)**: The price tag becomes too small and poorly illuminated at night, making it unreadable and leading to text recognition failure.
> > > >
> > > >
> > > > * **Spatial Confusion from Limited View (Navigation)**: Restricted field of view at night hides key spatial cues (e.g., landmarks, corridor orientation), causing incorrect navigation decisions.
> > > >
> > > >
> > > > * **Motion Blur (Action Recognition)**: Fast hand movement introduces motion blur, making the model misinterpret the content displayed on the screen.
> > > >
> > > > We will add these in-depth results and analysis into the revised version.
> > > >
> > > > # Ethics Concerns:
> > > > Thank you for raising this question. EgoNight-Oxford is derived from the publicly available Oxford Day-and-Night dataset, which is distributed under the BSD-3-Clause license (as stated on its HuggingFace repository). This license explicitly permits redistribution and modification, including creating derivative subsets, as long as the original copyright notice, license text, and disclaimer are retained.  Therefore, we have the legal and ethical right to redistribute the modified subset used in EgoNight-Oxford.  We will also appropriately acknowledge the original dataset and include the proper reference in our distribution.
> > > >
> > > > # W4: Presentation Can Be Further Addressed.
> > > > We thank the reviewer for pointing out issues related to phrasing, grammar, and clarity. We have carefully revised the paper to improve readability and presentation quality. This includes correcting grammatical errors (Page 2, L83-84), simplifying convoluted sentences (Page 2, L70-71), and refining the descriptions of the challenge (Page 2, L74-75). All changes are marked in the updated PDF (highlighted in red color). We further improve the clarity of the writing for the data generation in Sec 3.1, and modify the spacing at beginning of the paragraphs according to the suggestion

---

> > > > > ### Author Response · Authors · 2025-11-26
> > > > >
> > > > > Dear Reviewer,
> > > > >
> > > > > Understanding your busy schedule, we wanted to check if the revisions and details provided have addressed your concerns. Please let us know if any issues remain or further clarification is needed. Your feedback is invaluable to us for improving our work.
> > > > >
> > > > > Thank you for your attention.

---

> > > > > > ### Author Response · Authors · 2025-11-28
> > > > > >
> > > > > > Considering that the rebuttal discussion phase for ICLR is about to end, we sincerely hope to receive your further feedback so that we can continue our discussion. Once again, thank you for your hard work and selfless help.

---

### Official Review · Reviewer_akCw · 2025-11-03

**Soundness:** 3
**Presentation:** 3
**Contribution:** 3
**Rating:** 6
**Confidence:** 4

**Summary:**

The paper introduces EgoNight, a benchmark focused on egocentric vision understanding in nighttime. It consists of three data sources: (i) EgoNight-Synthetic, 50 ideally aligned egocentric pairs with varying illumination levels using a simulator; (ii) EgoNight-Sofia, 20 pairs of real-world egocentric videos with spatially and temporally aligned day–night counterparts; (iii) EgoNight-Oxford,  20 nighttime videos from the Oxford Day-and-Night dataset

**Strengths:**

- While most egocentric datasets are overwhelmingly focused on well-lit, daytime scenarios, this paper provides valuable nighttime egocentric data and thus has high significance for real-world applications.
- The proposed benchmark includes a comprehensive list of QA tasks, along with three critical tasks (day-night correspondence retrieval, temporal localization, egocentric depth estimation) that are unique for nighttime videos.
- The paper is well-written and easy to follow with carefully designed figures.

**Weaknesses:**

- The EgoNight-Sofia (real-world) dataset is the most valuable and novel component in the proposed benchmark. However, it is quite small with only 20 video pairs. The vast majority of QA pairs (2029 of 3658) come from the synthetic dataset, which may not capture the full complexity of real-world sensor noise and lighting artifacts.
- The day-augmented annotation pipeline could introduce a bias. It risks generating questions about details that are perfectly visible in the day video but are genuinely invisible or non-inferable in the night video. Is this considered in the human verification stage?
- The evaluation uses an "LLM-as-a-Judge" (GPT-4.1) to score answers, which might introduce bias towards its own outputs. GPT4.1 achieving the highest accuracy under this metric is less convincing.

**Questions:**

- Could the authors explain the human verification process for the day-augmented annotation? Specifically, how were QA pairs handled if the answer was only derivable from the day video and not the night video?
- Are there any strategies to address the synthetic-to-real gap in EgoNight-Synthetic?
- In Figure5, is NonC in the egoNight-Syn bar plot a typo? Could the authors use the same color for the same task to improve readability?
- What is the performance of reasoning on MLLM-generated captions of the videos?

---

> ### Author Response · Authors · 2025-11-22
> **Part 1: Comment about W1 , W2**
>
> Thank you for the constructive feedback and positive assessment of the benchmark. We address each concern below.
>
> # W1: Size of EgoNight-Sofia and the Quality of EgoNight-Synthetic
> We acknowledge that EgoNight-Sofia contains only 20 day–night video pairs; however, as a testbed, this scale is sufficient and comparable to existing egocentric benchmarks.
>
> * **Dataset Range** In practice, the usefulness of a benchmark is determined by the number and diversity of VQA pairs, not merely the count of raw videos. For example, as summarized below, EgoLife includes only 6 videos and EgoTempo 40, while EgoNight-Sofia/EgoNight provide 20/90 videos and 813/3658 VQA pairs, within a practical range.
> | Benchmark              | Conference | # Videos | # VQA Pairs |
> | ---------------------- | ---------- | -------- | ----------- |
> | EgoLife                | CVPR 2025  | 6        | 6,000       |
> | EgoTempo               | CVPR 2025  | 40       | 500         |
> | EgoNight-Sofia (Ours)  | –          | 20       | 813         |
> | EgoNight-Oxford (Ours) | –          | 20       | 816         |
> | EgoNight (Ours)        | –          | 90       | 3,658       |
>
> * **Scene converage** As described in Fig. 1, Sec. 3.1, and Sec. A.1, EgoNight-Sofia scenes were carefully recorded and intentionally varied (apartments, offices, streets, neighborhoods, tourist sites, grocery stores, outdoor fitness areas), ensuring broad coverage and natural diversity of VQA annotations.
> * **Performance spread** Tab. 1 shows a wide performance spread (≈10–40%), confirming that EgoNight-Sofia is sufficiently challenging and effective for distinguishing model capabilities.
>
> Regarding the proportion of synthetic vs. real data and the quality of synthetic data:
>
> * **synthetic-real balance** The number of videos across the three subsets—Synthetic (50), Sofia (20), and Oxford (20)—is proportionally reflected in the VQA annotations (2029 : 813 : 816). This results in an approximately 1:1 balance between synthetic and real (Sofia + Oxford) VQA samples, ensuring that our benchmark is not dominated by synthetic content.
>
> * **High Quality and Realism** EgoNight-Synthetic is built using high-fidelity 3D simulation (Sec. A.1; Fig. 6 & Fig. 9) with three difficulty levels that simulate realistic nighttime challenges, including illumination reduction and motion blur. Model performance in Table 1 consistently degrades with increasing difficulty, demonstrating that the synthetic data successfully captures real-world nighttime degradation patterns.
> * **Statistical Correlation Between Synthetic and Real Performance** We compute Pearson correlations between model performance on synthetic and real subsets (Appendix Tables 9–11):
>
>         Synthetic ↔ Sofia: 0.9359 (p = 6.847×10⁻⁵)
>
>         Synthetic ↔ Oxford: 0.8588 (p = 0.001462)
>
>     These strong and statistically significant correlations indicate that performance on synthetic data is highly predictive of performance on real-world data, further validating its representativeness.
>
> * **Synthetic Data Domain Transfer** When Qwen2.5-VL-7B is fine-tuned on synthetic data only, real-world evaluation (Sofia + Oxford) improves by ~6%. This confirms that EgoNight-Synthetic is not only a reliable testbed but also a valuable training source—demonstrating successful synthetic-to-real transfer.
> | Model                       | Object | Text  | Spatial | SceneSeq | Navigation | Light | Counting | Average |
> | --------------------------- | ------ | ----- | ------- | -------- | ---------- | ----- | -------- | ------- |
> | Qwen2.5-VL-7B               | 7.78   | 10.81 | 21.49   | 18.58    | 18.18      | 1.52  | 15.38    | 14.83   |
> | Qwen2.5-VL-7B-Synthetic-SFT | 14.44  | 24.32 | 23.43   | 20.18    | 26.26      | 20.00 | 16.15    | 20.57   |
>
> * Finally, EgoNight-Synthetic is the only subset with perfect day–night alignment except illumination, enabling controlled evaluation of illumination robustness, which is unachievable with real videos.
>
> # W2: Potential Bias in the Day-Augmented Annotation Pipeline
> Thank you for raising this concern. We kindly clarify that there might be a misunderstanding of our pipeline. As described in Fig. 2 and Sec. 3.2, **questions are always generated from the nighttime video, while the paired daytime video is used only to assist in producing accurate answers.** Therefore, the scenario in which the system “generates questions about details visible only in the day video but invisible in the night video” cannot occur by design: objects or events not visible in the night video would not be used to form questions.
>
> Our human verification further guards against this. As detailed in Fig. 7, Sec. A.2.1 (Appendix),  the annotators are required to make sure the same QAs apply to both day and night videos for those paired QA types. QA pairs where the answer is only derivable from the daytime video are rejected or corrected.  Together, these steps ensure that the final QA set is unbiased.

---

> ### Author Response · Authors · 2025-11-22
> **Part 2: Comment about W3, Q1, Q2, Q3**
>
> # W3: Concern on Bias from “LLM-as-a-Judge”
>
> Thank you for highlighting this important concern. We acknowledge that LLM-based evaluation is not perfectly unbiased. However, it is becoming a widely adopted and accepted practice in egocentric VQA, particularly for open-ended questions. Recent benchmarks such as EgoCross, EgoTempo, and EgoLife all rely on LLM-as-a-Judge for evaluating free-form answers.
>
> * **Human Verification Reduces GPT-Induced Bias** To minimize potential stylistic or factual bias from GPT-generated content, every QA pair in EgoNight is rigorously human-verified (Sec. A.2.1). Annotators inspect, correct, or rewrite both questions and answers using a structured protocol.
>
>    * Each video typically requires 1.5–2 hours of manual annotation.
>
>    * The human modification rate is high (~60%), indicating substantial intervention beyond GPT output.
>
>   This process significantly reduces GPT-specific linguistic artifacts and mitigates evaluation bias.
>
> * **Does GPT-4.1 Favor GPT-Style Answers?** We divided all QA annotations into two groups:
>
>    (a) Answers verified but not modified by humans (GPT-like style)
>
>    (b) Answers modified or fully created by humans (human style)
>
>
>    | Group                   | Portion | GPT-4.1 Evaluation Accuracy |
>    | ----------------------- | ------- | --------------------------- |
>    | (a) GPT-style answers   | ~40%    | 26.73%                      |
>    | (b) Human-style answers | ~60%    | 27.87%                      |
>
>    The scores are comparable, showing no clear preference for GPT-style wording.
>    This suggests that LLM-as-a-Judge evaluates semantic correctness rather than stylistic similarity, reducing concerns of bias toward GPT-like expressions.
>
> * **Is LLM-as-a-Judge reliable relative to humans?** We randomly sampled 300 QA pairs (questions, ground truth, model answers, and the corresponding LLM-assigned scores) and asked human evaluators to judge whether each score from the LLM was correct. This yielded an agreement rate of 95.67%, indicating strong alignment between human judgment and the LLM-as-a-Judge decisions, and thus demonstrating the reliability of the LLM-based evaluation. We added this part to the appendix page 21 Sec. A.5.1 line 1130-1137. And add reference from the main paper at page7 Sec. 4.1 line 367 .
>
> # Q1: Details about the Human Verification Period
> Thank you for the question.  We provide the detailed annotation instructions in Fig.7, Sec. A.2.1, Appendix. In addition to the written guidelines, we provided an onboarding meeting to make sure the annotators understand the criteria of annotation and to clarify questions that might come up.  Briefly, for each VQA pair, annotators are required to delete/modify/add operations, and as answered in W2, we ask the annotators to ensure consistency between day and night pairs.
>
> # Q2: Strategies to Address Synthetic-to-Real Gap
> Thank you for the question. EgoNight-Synthetic was designed with several measures to reduce the synthetic-to-real gap. As described in Sec. A.1, we use the Blender Cycles physically based renderer and apply multiple strategies to approximate real nighttime conditions, including:
>
> * Scene refinement: manual editing to remove unrealistic elements and create natural layouts.
> * Illumination diversity: adding or adjusting light sources and varying light types (e.g., spotlights to mimic mobile flashlights).
> * Human-like camera motion: manually planned trajectories to simulate egocentric exploration.
> * Realistic nighttime degradation: simulating sensor noise (via low sampling rate), varying lighting strength/coverage, and adding motion blur through shutter-speed adjustments.
>
> The visual fidelity (Fig. 6, Fig. 9, Appendix) and the consistent ranking of models across synthetic and real subsets (Tab. 1) indicate that the synthetic data captures challenges similar to real nighttime videos. Moreover, as shown in W1, fine-tuning on EgoNight-Synthetic improves performance on real data, further demonstrating its effectiveness in bridging the synthetic-to-real gap.
>
> # Q3: Typo and Plot
> Thank you for pointing this out. “NonC” is not a typo, it is short for non–common sense reasoning. We have clarified this explicitly in the figure caption to avoid confusion. As suggested, we have also updated Figure 5 so that bars corresponding to the same task use the same color.

---

> > ### Author Response · Authors · 2025-11-22
> > **Part 3: Comment about Q4**
> >
> > # Q4: Caption Reasoning Performance
> > We use MLLM-generated captions as intermediate guidance during question generation. Specifically, during this stage, the MLLM receives both the captions and video frames, ensuring that the generated questions are consistent with the captioned semantics while remaining grounded in the visual content. To further improve caption quality, we employ task-specific prompting as shown in Fig. 8 (Appendix).
> > Quantitatively evaluating “caption reasoning performance” is non-trivial because this step serves as an auxiliary component rather than a standalone prediction task. We have not yet identified a reliable automatic metric for this. As an alternative, we will include qualitative examples in the Appendix to illustrate that MLLMs can effectively perform this meta-task and produce captions that support high-quality question generation.
> >
> > We have revised the paper according to your suggestions (page7 Sec. 4.1 line 367, Sec. A.5.1, Fig. 5; highlighted in blue color). We would greatly appreciate it if you could let us know whether our responses adequately address your concerns, and whether the revisions resolve the issues you raised.

---

> > > ### Author Response · Authors · 2025-11-26
> > >
> > > Dear Reviewer,
> > >
> > > Understanding your busy schedule, we wanted to check if the revisions and details provided have addressed your concerns. Please let us know if any issues remain or further clarification is needed. Your feedback is invaluable to us for improving our work.
> > >
> > > Thank you for your attention.

---

> ### Comment · Reviewer_akCw · 2025-11-27
>
> Thanks for the responses! I think this paper fills an important gap by providing nighttime egocentric data, and I maintain my positive rating. However, I think it's unfair to compare the number of videos to EgoLife, since its videos are extremely long (300 hours total), while most EgoNight videos are within 80s (shown in figure 4).
> - (minor) In Figure 5, EgoNight-Synthetic includes “NonC” while EgoNight-Sofia includes “Action” (with the remaining 11 QA types being identical). Could you explain why the QA types differ between these two subsets?

---

> ### Author Response · Authors · 2025-11-28
>
> Thank you for your feedback! We acknowledge that EgoLife is a long-duration video dataset. Our intention in including it was to illustrate that a dataset’s value is not solely determined by the number of videos, but also by the number and quality of VQA pairs, which are crucial for benchmarking. To provide a fairer comparison, we have also referenced other short-duration egocentric datasets, such as EgoTempo (40 videos, 500 VQA pairs, ~45s per video).
>
> Furthermore, we would like to emphasize our contribution during the data collection. Because in EgoNight, each nighttime video has a perfectly aligned daytime counterpart, forming paired data (20 night time videos in EgoNight-Sofia, but 40 videos in total). Which means we need to collect these data by ourself instead of annotating video from Ego4D or other datasets as EgoTempo and EgoCross did. This effectively doubles (or even more due to the alignment) the recording duration for each pair, providing richer temporal and environmental variation compared to single standalone videos.
>
> Regarding the QA types, “Non-common” appears only in the synthetic subset because it is designed to detect rare or unusual cases that occur in synthetic environments but are unlikely in real-world data. Conversely, the “Action” QA type is relevant only to real-world scenarios, where human activities are naturally present.
>
> We sincerely thank you for your positive feedback and constructive suggestions on our paper, which has greatly motivated us. We hope these clarifications help further address your concerns and demonstrate the value of our contributions. We would be grateful if these revisions positively inform your overall assessment, and we would be happy to address any additional questions or suggestions you may have. Thanks again.

---

### Author Response · Authors · 2025-11-27

Dear Reviewers and Area Chairs,

We sincerely appreciate all reviewers for their insightful and constructive feedback. According to these comments, we have improved the paper and corresponding supplementary materials (new pdf uploaded) and highlighted the main changes with blue, red, and brown text. Below, we summarize all changes:
1. We supplement concern about size of real dataset L404-405 denoted as blue color, and Appendix A.8.1 L1510-L1595 (reviewer akCw,  Cyx7, 3vts)

2. We supplement in-depth analysis about how to enhace model performance in L108-109, L456-482, Table 2, Table 3 denoted as red color. And more failure case analysis in Appendix A.9 L1499-1506 A.7.1 L1355-1375 (reviewer Cyx7, VX41)

3. We supplement results of synthetic data quality analysis, and cross-domain fine-tuning improvement in L107-108, L294-295, L429-455, denoted as brown color. (reviewer akCw, Cyx7, 3vts)

4. We supplement concern about potential bias of GPT4.1 and validate LLM-as-a-Judge strategy in L377 denoted as blue color, and Appendix A.5.1 L1184-L1187 (reviewer akCw, Cyx7)

5. We supplement ethic statement in Appendix L1617-1623. (reviewer Cyx7,  VX41)

6. We supplement concern about balance between synthetic and real data in L307-L309 denoted as blue color (reviewer akCw)

7. We supplement more inter-annotator statistics in Appendix L917-921 (reviewer VX41)

8. We supplement qualitative result of caption reasoning in Appendix Figure. 13, (reviewer akCw)

9. We supplement confusions, typos, presentation problems in Figure 5, L76-77, L70-71, L74-75, and we improve the clarity of the writing for the data generation in Sec 3.1(reviewer akCw, Cyx7)

---

### Meta-Review · Area_Chair_6Tbv · 2026-01-07

**Summary:**

EgoNight is the first comprehensive benchmark designed to evaluate egocentric vision understanding in low-light and nighttime conditions—a critical gap in current research. The suite features a diverse dataset comprising synthetic scenes (EgoNight-Synthetic), real-world aligned day-night pairs (EgoNight-Sofia), and unaligned nighttime footage (EgoNight-Oxford). By leveraging day-night alignment, the authors developed a "day-augmented auto-labeling" pipeline to generate 3,658 human-verified QA pairs across 12 categories, alongside auxiliary tasks like correspondence retrieval and depth estimation. Evaluations of state-of-the-art Multimodal Large Language Models (MLLMs), including GPT-4.1 and Gemini, reveal a significant performance drop in nighttime scenarios, highlighting a lack of robustness in current systems and establishing EgoNight as a vital tool for developing more reliable AI assistants.


The paper is recognized for its strong motivation and the effort involved in data collection. However, the reviewers raised the concern on the incremental novelty, and the concerns of "LLM-as-a-judge" bias and clarify how the small-scale real-world data provides enough statistical power to generalize the findings. The main pros and cons highlighted by the reviewers are listed below:


**Pros:**
* High Task Significance: The paper addresses a critical, under-explored gap in egocentric vision—low-light and nighttime understanding. Reviewers unanimously agree that this is a timely contribution with high practical relevance for the development of robust personal AI assistants.
* Methodological Rigor in Alignment: A key technical strength is the creation of the EgoNight-Sofia dataset. The use of a "video-guided recording protocol" to produce temporally and spatially aligned day-night pairs provides a robust foundation for isolating illumination as a controlled variable. This, combined with high-fidelity synthetic data, allows for a nuanced analysis of model failure modes.


**Cons:**
* Limited Technical Novelty: Reviewer Cyx7 noted that the work follows an established VQA paradigm, framing it as a domain-specific application (nighttime) rather than a fundamental algorithmic breakthrough. The contribution is seen as more empirical and resource-oriented than theoretical.
* Concerns over Dataset Scale: While the alignment protocol is praised, the scale of the real-world data (EgoNight-Sofia) is notably small, consisting of only 20 pairs, which is raised by multiple reviewers. There are concerns that the benchmark’s reliance on synthetic data may limit its ability to reflect the complex sensor noise and artifacts inherent in real-world nighttime photography.
* Potential for Evaluation Bias and Contamination: The methodology relies heavily on GPT-4.1 for both question generation and as a "judge" for evaluation. This raises significant concerns regarding stylistic bias and "dataset contamination," where the benchmark might unfairly favor models that share an architectural or training lineage with GPT-4.1.

**Reviewer Concerns:**

The authors actively participated in the rebuttal period and added more detailed clarification and additional experiment results. These additional information clarified most of the concerns like dataset scale and potential evaluation bias.

There are still skeptical on the significance of the contribution and the scales.

**Reviewer Scores:**

Reviewer akCw remains positive score after rebuttal.

Reviewer Cyx7 will likely to remain the score.

Reviewer 3vts commented he leaning towards positive after rebuttal, so likely to change the score from 4 to 6.

Reviewer VX41 commented that he increased the score, so will be from 6 to 8.

---

### Decision · Program_Chairs · 2026-01-26

Accept (Poster)